# C11orf54 promotes DNA repair via blocking CMA-mediated degradation of HIF1A

Junyang Tan [1,2,3], Wenjun Wang[1,2,3], Xinjie Liu[1,2], Jinhong Xu[1,2], Yaping Che[1,2], Yanyan Liu[1,2], Jiaqiao Hu[1,2], Liubing Hu[1,2], Jianshuang Li [1,2,4 ✉] & Qinghua Zhou [1,2,4 ✉]

C11orf54 is an ester hydrolase highly conserved across different species. C11orf54 has been identified as a biomarker protein of renal cancers, but its exact function remains poorly understood. Here we demonstrate that C11orf54 knockdown decreases cell proliferation and enhances cisplatin-induced DNA damage and apoptosis. On the one hand, loss of C11orf54 reduces Rad51 expression and nuclear accumulation, which results in suppression of homologous recombination repair. On the other hand, C11orf54 and HIF1A competitively interact with HSC70, knockdown of C11orf54 promotes HSC70 binding to HIF1A to target it for degradation via chaperone-mediated autophagy (CMA). C11orf54 knockdown-mediated HIF1A degradation reduces the transcription of ribonucleotide reductase regulatory subunit M2 (RRM2), which is a rate-limiting RNR enzyme for DNA synthesis and DNA repair by producing dNTPs. Supplement of dNTPs can partially rescue C11orf54 knockdown-mediated DNA damage and cell death. Furthermore, we find that Bafilomycin A1, an inhibitor of both macroautophagy and chaperone-mediated autophagy, shows similar rescue effects as dNTP treatment. In summary, we uncover a role of C11orf54 in regulating DNA damage and repair through CMA-mediated decreasing of HIF1A/RRM2 axis.

[1] The Sixth Affiliated Hospital of Jinan University, Jinan University, 523573 Dongguan, Guangdong, China. [2] The Biomedical Translational Research Institute, Health Science Center (School of Medicine), Jinan University, 510632 Guangzhou, Guangdong, China. [3] These authors contributed equally: Junyang Tan, Wenjun Wang. [4] These authors jointly supervised this work: Jianshuang Li, Qinghua Zhou. ✉email: lijianshuan1314@jnu.edu.cn; gene@email.jnu.edu.cn

Genomic integrity is crucial in homeostasis maintenance, normal development, and cancer prevention[1]. Both endogenous and exogenous genomic stress cause DNA single-strand breaks (SSBs) and DNA double-strand breaks (DSBs), leading to active DNA damage response (DDR)[2,3]. DDR pathways sense, signal, and repair different types of DNA damage, which is crucial for maintaining genomic stability[4]. Homologous recombination (HR) represents the primary mechanism for the error-free homology-directed repair of DSBs and interstrand crosslinks (ICLs). The Rad51 and its paralogs play essential roles in this process[5,6]. Ribonucleotide reductase (RNR) catalyzes deoxyribonucleotides (dNTP) needed for DNA synthesis. The two subunits RRM1 (Ribonucleotide Reductase Catalytic Subunit M1) and RRM2 (Ribonucleotide Reductase Regulatory Subunit M2) constitute the α2β2 complex to exert catalytic activity, and the balance of dNTPs within cells is vital for cellular homeostasis and maintaining genomic integrity[7–9]. The rate-limiting RNR enzyme, RRM2 is essential for DNA synthesis and DNA repair by producing dNTPs[10].

Hypoxia-inducible factor-1 (HIF1) is a dimeric transcriptional complex that participates in many biological processes such as metabolism, inflammation, and tumorgenesis[11]. HIF1A is a subunit of the HIF1 complex, which was modified by HIF-specific prolyl-hydroxylases (PHDs) in the presence of $O_2$. Modified HIF1A is degraded by the proteasome, which requires oxygen-dependent proline hydroxylation and von Hippel-Lindau (VHL)-mediated ubiquitination[12]. Furthermore, HIF1A was reported to regulate DDR in various ways. P-H2A.X is a marker for DNA double-strand breaks to amplify the damage signal and recruits DNA damage repair proteins[13]. In cancer cells, knockdown of HIF1A reduces hypoxia-induced p-H2A.X accumulation and then affects the capacity of DNA damage repair and tumor therapy resistance[14]. Hypoxia-mediated stabilization of HIF1A promotes the transcription of hTERT (telomerase reverse transcriptase) and hTR (telomerase RNA component) and then regulates DNA damage and genomic stability[15,16]. Meanwhile, loss of HIF1A could restrain DNA double-strand break repair and be more sensitive to chemotherapy[17,18].

Chaperone-mediated autophagy (CMA) is a type of autophagy in which substrates are directly targeted to the lysosome for degradation. CMA substrate proteins are selectively recognized by heat-shock cognate protein of 70 kDa (HSC70) and targeted to lysosomes, subsequently recruited to lysosomal membrane receptor type 2A (LAMP2A) for CMA-mediated degradation[19]. CMA participates in DNA repair and may prevent malignant transformation by maintaining genome stability[19]. Recent studies demonstrated that HIF1A is a CMA substrate and is involved in cell cycle regulation. HIF1A is degraded in a CMA-dependent manner when it is ubiquitylated on Lys63 by the E3 ubiquitin-protein ligase STUB1[20]. Checkpoint kinase 1 (CHK1), the cell cycle regulator, is another CMA substrate. CMA inhibition causes nuclear persistence of Chk1 during DNA damage response, leading to DNA damage accumulation and impairing DNA repair[21].

*C11orf54* (chromosome 11 open reading frame 54), also known as *PTD012*, is a highly conserved gene expressed in *Mus musculus*, *Brachydanio rerio*, *Drosophila melanogaster*, and *Caenorhabditis elegans*[22]. Human C11orf54 contains a $Zn^{2+}$ ion coordinated to three histidine residues and exhibits ester hydrolase activity[22]. In addition, the *Drosophila melanogaster* ortholog of C11orf54 plays a critical role in protein homeostasis during overnutrition[23]. C11orf54 was identified as a biomarker protein of endometrial cancer, renal cell carcinoma, and clear cell renal cell carcinoma by two-dimensional gel electrophoresis coupled with mass spectrometry[24–26]. However, its function in mammals is unclear yet.

In the present study, we found that C11orf54 regulated cell proliferation and apoptosis. Knockdown of C11orf54 activated ATM-dependent DNA damage response (DDR) and inhibited homologous recombination repair by repressing the expression of Rad51. Furthermore, we found that C11orf54 competitively interacted with HSC70, which interrupted the interaction between HIF1A and HSC70. Thus, knockdown of C11orf54 enhanced the binding of HSC70 to HIF1A, leading to the degradation of HIF1A via chaperone-mediated autophagy. Furthermore, blocking CMA by bafilomycin A1 (BafA1) could partially recover C11orf54 knockdown-induced DNA damage and cell proliferation suppression. Our data demonstrated that C11orf54 promotes DNA repair via blocking CMA-mediated degradation of HIF1A.

## Results

**C11orf54 is predominantly localized in the cytoplasm.** The crystal structure of C11orf54 revealed it as an ester hydrolase, which might belong to the superfamily of metallo-β-lactamase fold proteins[22]. C11orf54 is conserved among species, including *Homo sapiens*, *Mus musculus*, *Brachydanio rerio*, *Drosophila melanogaster*, and *Caenorhabditis elegans*. C11orf54 expresses universal among different tissues in human species, especially enriched in kidney and liver tissues (Supplementary Fig. 1). Thus, we used 293T and hepatocytes to investigate the physiological function of C11orf54.

A previous study demonstrated that C11orf54 is predominantly located in nuclear by using an automatic phenotyping approach[27]. In another study, C11orf54 was detected in the cytoplasm and nucleus in C11orf54 overexpressed cells[28]. To study the exact subcellular localization of endogenous C11orf54, we performed the nuclear/cytosolic-fractionation assay and immunostaining experiment using the verified antibodies. First, we generated C11orf54 knockdown cells using two different shRNAs (shC11orf54-1, shC11orf54-2)-mediated gene silencing. The protein and mRNA levels of C11orf54 were significantly reduced in C11orf54 knockdown cells (Fig. 1a, b). Then we overexpressed C11orf54 in the knockdown cells to check the specificity of the C11orf54 antibodies (Fig. 1c). The absence of blots indicated a successful construction of C11orf54 knockdown cell line and also validated the specificity of C11orf54 antibody. The immunostaining experiments showed that endogenous C11orf54 was predominantly localized in the cytoplasm (Fig. 1d, e). Moreover, the nuclear/cytosol fractionation assay showed consistent results that C11orf54 was primarily observed in the cytoplasm fractionation (Fig. 1f, g). These data collectively suggest that C11orf54 is predominantly localized in the cytoplasm.

**C11orf54 deficiency suppresses cell proliferation and promotes apoptosis.** Next, we investigated whether knockdown of C11orf54 affects cell proliferation and apoptosis, which are key cellular events in the development of organisms[29]. CCK8 assay showed that C11orf54 silencing significantly inhibited the PLC/PRF/5 and 293T cell growth (Fig. 2a; Supplementary Fig. 2a) and colony formation (Fig. 2b, c and Supplementary Fig. 2b, c). Furthermore, we detected a significantly reduced number of EdU-positive cells in the C11orf54 knockdown cell compared to the control cell (Fig. 2d, e). These results indicate that C11orf54 deficiency suppresses cell proliferation.

Interestingly, we found that the expression of C11orf54 has increased upon Cisplatin (CDDP) treatment in a time-dependent manner (Fig. 2f, g). To examine whether other genotoxic inducers could regulate the expression of C11orf54, we evaluated C11orf54 expression upon Hydroxyurea (HU) and Camptothecin (CPT) treatment. Similarly, HU and CPT could also promote the expression of C11orf54 in a time-dependent

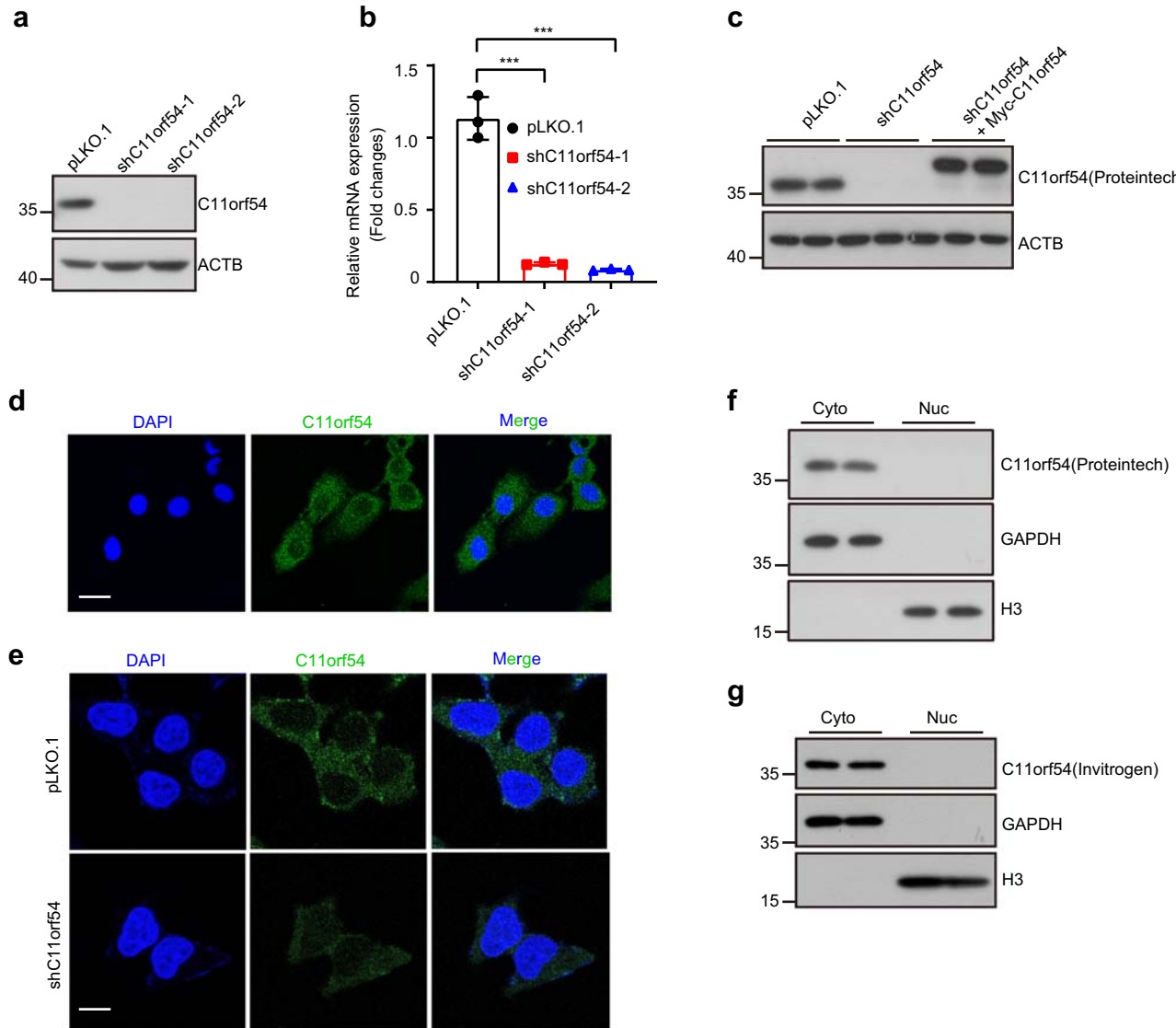

**Fig. 1 C11orf54 is predominantly localized in the cytoplasm. a, b** Western blot (**a**) and qPCR (**b**) experiments indicate the knockdown efficiency of C11orf54 (shC11orf54-1, shC11orf54-2) in PLC/PRF/5 cell line. ACTB was used as a control (n = 3 biological replicates, data are presented as mean values ± SD, ***p < 0.001). **c** Western blot indicates the specificity of the C11orf54 antibody in the C11orf54 knockdown cells and reintroducing C11orf54 in the C11orf54 knockdown cells. **d** Representative immunostaining images confirm the subcellular localization of C11orf54 in wild-type cells. Scale bar = 20 µm. **e** Representative immunostaining images confirm the subcellular localization of C11orf54 in control and C11orf54 knockdown PLC/PRF/5 cells. Scale bar = 10 µm. **f, g** Nuclear/cytosol fractionation assay shows subcellular localization of C11orf54 in PLC/PRF/5 cell line with two verified antibodies from Proteintech (**f**) and Invitrogen (**g**). GAPDH was used as a cytoplasm (Cyto) marker and H3 as a marker of the nucleus (Nuc).

manner (Supplementary Fig. 3a–d). These data showed C11orf54 could respond to the genotoxic inducers, indicating that C11orf54 may play a role in the genotoxic drug-induced cell death. Indeed, the cell viability and colony formation assays showed that C11orf54 knockdown cells were more sensitive to cisplatin than control cells (Fig. 2h–j). Annexin V(FITC)/Propidium iodide (PI) double staining showed that Cisplatin-induced early and late phase apoptosis was increased in the C11orf54 knockdown cells (Fig. 2k, l). TUNEL assay also demonstrated that loss of C11orf54 promoted apoptosis under cisplatin treatment (Supplementary Fig. 3e, f). Moreover, we found that knockdown of C11orf54 enhanced the cleavage of caspase 3 and PARP1, which play central roles in apoptosis execution (Fig. 2m, n). Interestingly, the effect of C11orf54 on caspase3 activation seemed only in the presence of cisplatin, which may be due to cisplatin augmenting the apoptosis tendency after C11orf54 knockdown. Meanwhile, increased pro-apoptotic BAX and decreased anti-apoptotic Bcl-2 were observed

in the C11orf54 knockdown cells (Fig. 2m, n). Taken together, these results demonstrate that C11orf54 deficiency suppresses cell proliferation and promotes apoptosis.

**Knockdown of C11orf54 promotes DNA damage via suppression of homologous recombination.** Genotoxic drugs cause cell death by inhibiting DNA synthesis or breaking DNA structure. Our results showed that the protein level of C11orf54 was increased after the genotoxic drug treatment (Fig. 2f, g and Supplementary Fig. 3a–d), indicating that C11orf54 may be involved in genome stability or DNA damage. To examine whether C11orf54 participates in drug-induced DNA damage, we used a comet assay to detect double-strand breaks (DSBs) following cisplatin treatment. DNA fragments indicated by the tail-DNA and tail-moment were dramatically increased in C11orf54 knockdown cells after Cisplatin treatment (Fig. 3a–c).

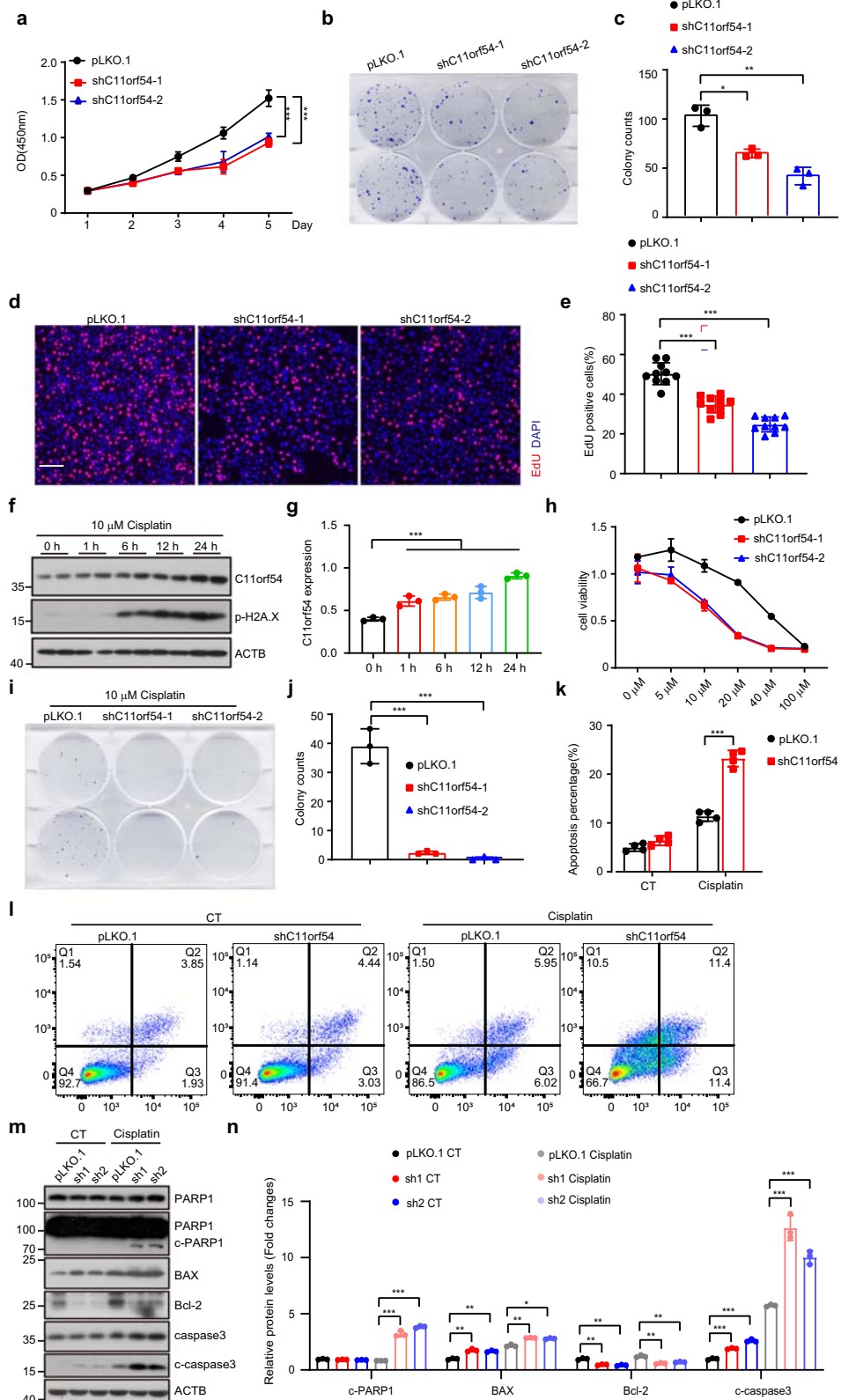

Additionally, we examined the γH2A.X (p-H2A.X-Ser139) level, which is recognized as a marker of DNA damage[30], by immunofluorescence staining and western blot assay. C11orf54 silencing significantly increased the number of γH2A.X foci (Fig. 3d, e) and protein level (Fig. 3f, g) under both cisplatin and normal conditions compared to the control cells. These results suggest that knockdown of C11orf54 promotes DNA damage.

ATM (ataxia-telangiectasia mutated) is one of the most upstream kinases in the DDR pathway, whose activation can trigger the phosphorylation of a series of kinases such as Chk1 and Chk2[31–34]. We found that knockdown of C11orf54 promoted the phosphorylation of ATM and its downstream Chk1/Chk2 under normal conditions and Cisplatin treatment (Fig. 3f, g). We could also observe the same phenomenon in 293T cells

**Fig. 2 C11orf54 knockdown suppresses cell proliferation and promotes apoptosis. a** CCK8 assay shows the cell survival of control and C11orf54 knockdown PLC/PRF/5 cells (data are presented as mean values ± SD, ***$p < 0.001$). **b, c** Colony formation (**b**) and quantitative results (**c**) show the cell growth of control and C11orf54 knockdown PLC/PRF/5 cells ($n = 3$ biological replicates; data are presented as mean values ± SD, *$p < 0.05$, **$p < 0.01$). **d, e** EdU staining confirms the effect of C11orf54 knockdown on cell proliferation in PLC/PRF/5 cell line. The images display EdU staining (red color) merged with DAPI staining (blue color). (Scale bar = 100 μm; data are presented as mean values ± SD, ***$p < 0.001$). **f, g** Western blots (**f**) and quantitative results (**g**) of C11orf54 expression in PLC/PRF/5 cells treated with 10 μM cisplatin for different times (0 h, 1 h, 6 h, 12 h and 24 h) ($n = 3$ biological replicates; data are presented as mean values ± SD, ***$p < 0.001$). **h** Viability assay after incubation with cisplatin in control and C11orf54 knockdown PLC/PRF/5 cells at different concentrations (5 μM, 20 μM, 40 μM, and 100 μM). **i, j** Colony formation analysis of C11orf54 knockdown on cisplatin sensitivity after treatment with 3 μM cisplatin for 24 h and cultured in a drug-free medium for another 10 days ($n = 3$ biological replicates; data are presented as mean values ± SD, ***$p < 0.001$). **k, l** Analysis of apoptosis by Annexin V(FITC)/Propidium iodide (PI) double staining (L) and quantitative results (K) in control and C11orf54 knockdown cells after treatment with 10 μM cisplatin for 48 h ($n = 4$ biological replicates; data are presented as mean values ± SD, ***$p < 0.001$). **m, n** Western blots (**m**) and quantitative results (**n**) of the indicated proteins in control and C11orf54 knockdown PLC/PRF/5 cells after treatment with 10 μM cisplatin for 48 h ($n = 3$ biological replicates; data are presented as mean values ± SD, *$p < 0.05$, **$p < 0.01$, ***$p < 0.001$).

---

(Supplementary Fig. 2d). Furthermore, ATM kinase inhibitor KU-60019 could partially repress the increased expression of p-ATM, p-CHK1, p-CHK2 and p-H2A.X in C11orf54 knockdown cells both in control or Cisplatin treatment conditions (Fig. 3h, i). These data suggest that knockdown of C11orf54 activates DNA damage in an ATM-dependent pathway.

DNA repair is an evolutionarily conserved pathway that can repair DNA damage from endogenous or exogenous sources to maintain genomic integrity[35,36]. We wondered whether C11orf54 regulates DNA repair deficiency. Firstly, we detected the expression of Rad51 and Ku70, two critical proteins in homologous recombination (HR) and non-homologous end-joining (NHEJ), respectively. We found that knockdown of C11orf54 significantly repressed the expression of Rad51 but not Ku70 (Fig. 4a, b), and immunostaining experiments showed loss of C11orf54 inhibited Rad51 foci formation in nucleus (Fig. 4c, d), which indicated that C11orf54 may be involved in homologous recombination. Thus, we evaluated the homologous recombination repair activity in C11orf54 knockdown and control cells using HR reporter system. In this system, GFP is interrupted by the I-SceI site and the functional GFP can be restored by HR repair using the downstream GFP as the template after I-SceI expression[37]. We co-transfected pDR-GFP and pCBA-SceI into C11orf54 knockdown and control cells, and then recorded the GFP positive cells by flow cytometry. We found that the percentage of GFP-positive cells was significantly reduced in C11orf54 knockdown cells (Fig. 4e, f), indicating that knockdown of C11orf54 suppressed HR repair activity. Furthermore, compared to the control cells, C11orf54 knockdown cells showed a slower clearance of p-H2A.X foci after withdrawing cisplatin treatment (Fig. 4g, h). Collectively, these data suggest that knockdown of C11orf54 impairs homologous recombination repair, which enhances the DNA damage.

**Knockdown of C11orf54 suppresses HIF1A-mediated nucleotides synthesis.** To identify the potential mechanism of C11orf54 knockdown-mediated homologous recombination repair impairment, we analyzed the transcriptomes in C11orf54 knockdown and control cells by high-throughput RNA sequencing (RNA-Seq). We identified 708 significant differentially expressed genes (303 upregulated genes and 405 down-regulated genes) in C11orf54 knockdown cells, which were visualized by the volcano plot (Fig. 5a). We further performed functional analyses of DEGs (Differentially Expressed Genes) using the KEGG database. The top 20 significant downregulated KEGG pathways were listed in Fig. 5b. Glycolysis/gluconeogenesis and HIF1 signaling pathway were downregulated in C11orf54 knockdown cells (Fig. 5b). Consistently, GSEA results showed that the gene sets related to hypoxia (NES = −1.4, $p$-value = 0.005) and glycolysis-gluconeogenesis (NES = −1.5, $p$-value = 0.011) were enriched in

pLKO.1 group (Fig. 5c). HIF1A is a known transcription factor that regulates the transcription of glycolysis and glycolytic genes[38]. Thus, we evaluated the expression of HIF1 target genes and glycolysis/gluconeogenesis-associated genes in the C11orf54 knockdown cells. The qPCR results showed that most glycolysis-associated genes (*GLUT1, GLUT2, PGK1, ENO1, PDK3, LDHA,* and *PDHA1*), as well as gluconeogenesis-associated genes (*FBP1* and *PEPCK*), were significantly decreased in C11orf54 knockdown cells (Fig. 5d). Moreover, knockdown of C11orf54 significantly repressed the protein level of PKM2, HK2, LDHA, PFKP, and HIF1A (Fig. 5e, f). These data indicate that knockdown of C11orf54 suppresses the HIF1A signaling pathway.

Next, we wondered which downstream target of HIF1A was involved in DNA repair in C11orf54 knockdown cells. Although the interaction between glycolysis and DNA repair pathways remains unclear, several studies have suggested that glycolysis may maintain genome stability by providing essential metabolites for DNA metabolism[39]. For example, the pentose phosphate pathway (PPP) converts the glycolysis intermediate (glucose-6-phosphate) to ribose-5-phosphate for the synthesis of nucleotides and NADPH to reduce DNA damage[40,41]. Thus, we evaluated the NADPH/NADP$^+$ ratio (a well-known biomarker of PPP) and the expression of Glucose-6-phosphate dehydrogenase (the rate-limiting enzyme of PPP) in the C11orf54 knockdown and control cell. However, there is no difference in the NADPH/NADP+ ratio and G6PD expression between C11orf54 knockdown and control cells (Supplementary Fig. 4a, b), suggesting that knockdown of C11orf54 mediated DNA damage may not go through the glycolysis pathway.

To further investigate how C11orf54 regulates DNA damage repair, we measured the mRNA levels of genes involved in DNA repair. As shown in Fig. 6a, most transcript levels of candidate genes were unchanged except RRM2. Additionally, the protein level of RRM2 was significantly decreased in C11orf54 knockdown cells under both cisplatin and control conditions (Fig. 6b, c). RRM2, the ribonucleotide reductase small subunit, is essential for DNA synthesis and repair by producing dNTPs[42]. It has been reported HIF-1α/STAT3 signaling pathway could upregulate RRM2 transcriptional level[43,44]. Therefore, we determined whether the supplementation of nucleosides could rescue C11orf54 knockdown-induced DNA damage. The supplementation of nucleosides partially reduced γ-H2AX foci in the C11orf54 knockdown cells (Fig. 6d, e). Furthermore, the suppressed colony formation in C11orf54 knockdown cells was also partially restored by nucleoside supplementation (Fig. 6f, g). These results suggest that C11orf54 knockdown causes DNA damge by suppressing HIF1A /RRM2 axis.

**Knockdown of C11orf54 causes HIF1A degradation via chaperone-mediated autophagy.** Next, we investigated how

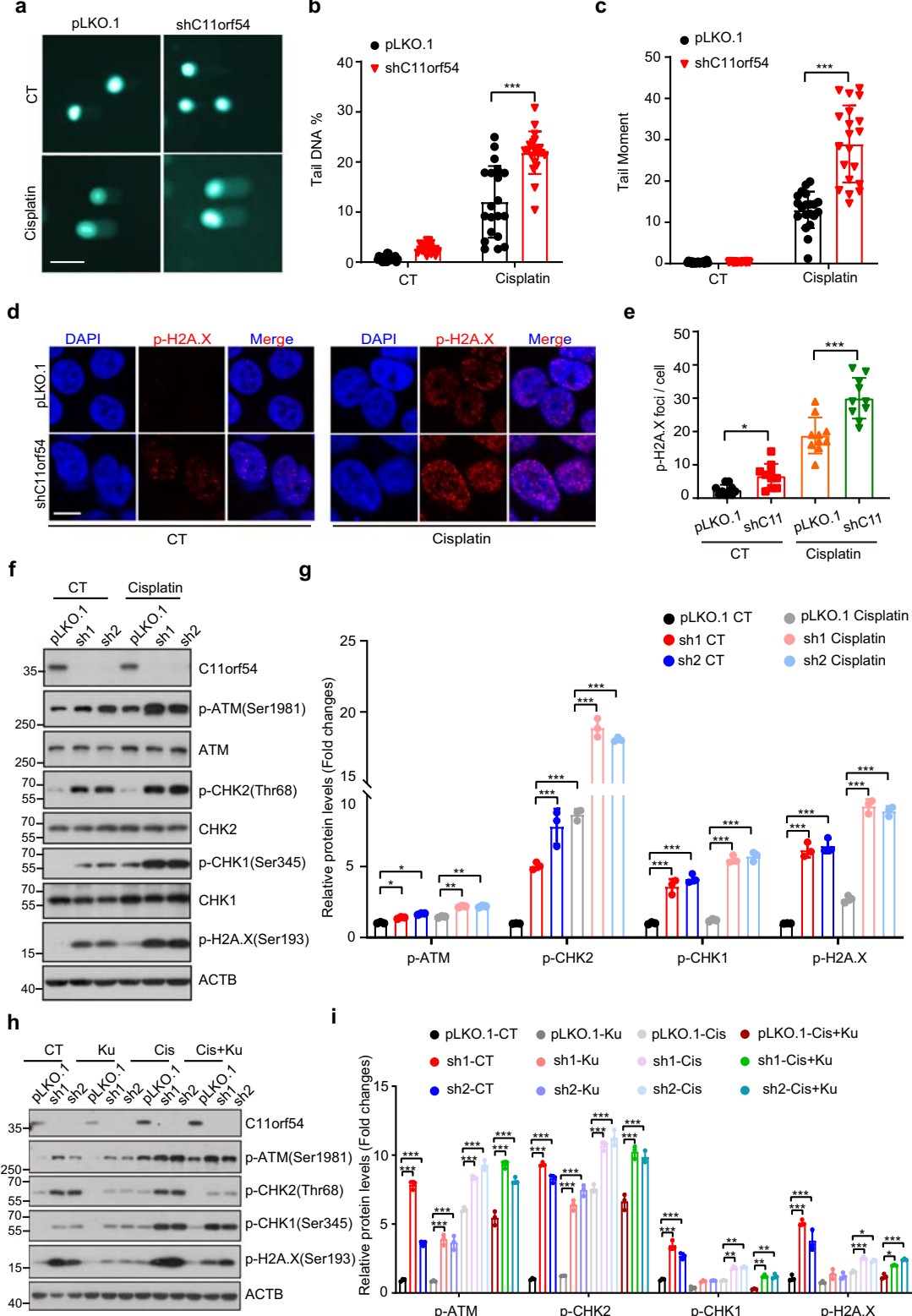

**Fig. 3 C11orf54 knockdown promotes DNA damage response. a–c** Representative images of comet assay in control and C11orf54 knockdown PLC/PRF/5 cells (**a**) and the quantification of tail DNA% (**b**) and tail moment (**c**) ($n = 20$ independent cells; data are presented as mean values ± SD; ***$p < 0.001$). Scale bar = 30 μm. **d, e** Representative images of γH2A.X foci (**d**) and quantitative results (**e**) in control and C11orf54 knockdown PLC/PRF/5 cells ($n = 10$ independent cells; data are presented as mean values ± SD; *$p < 0.05$, ***$p < 0.001$). Scale bar = 10 μm. **f, g** Western blots (**f**) and quantitative results (**g**) of the indicated proteins in control and C11orf54 knockdown PLC/PRF/5 cells upon 10 μM cisplatin treatment for 6 h ($n = 3$ biological replicates; data are presented as mean values ± SD; *$p < 0.05$, **$p < 0.01$, ***$p < 0.001$). **h, i** Western blots (**h**) and quantitative results (**i**) of the indicated proteins in control and C11orf54 knockdown PLC/PRF/5 cells upon 10 μM cisplatin and 10 μM cisplatin plus 10 μM KU-60019 for 6 h ($n = 3$ biological replicates; data are presented as mean values ± SD; *$p < 0.05$, **$p < 0.01$, ***$p < 0.001$).

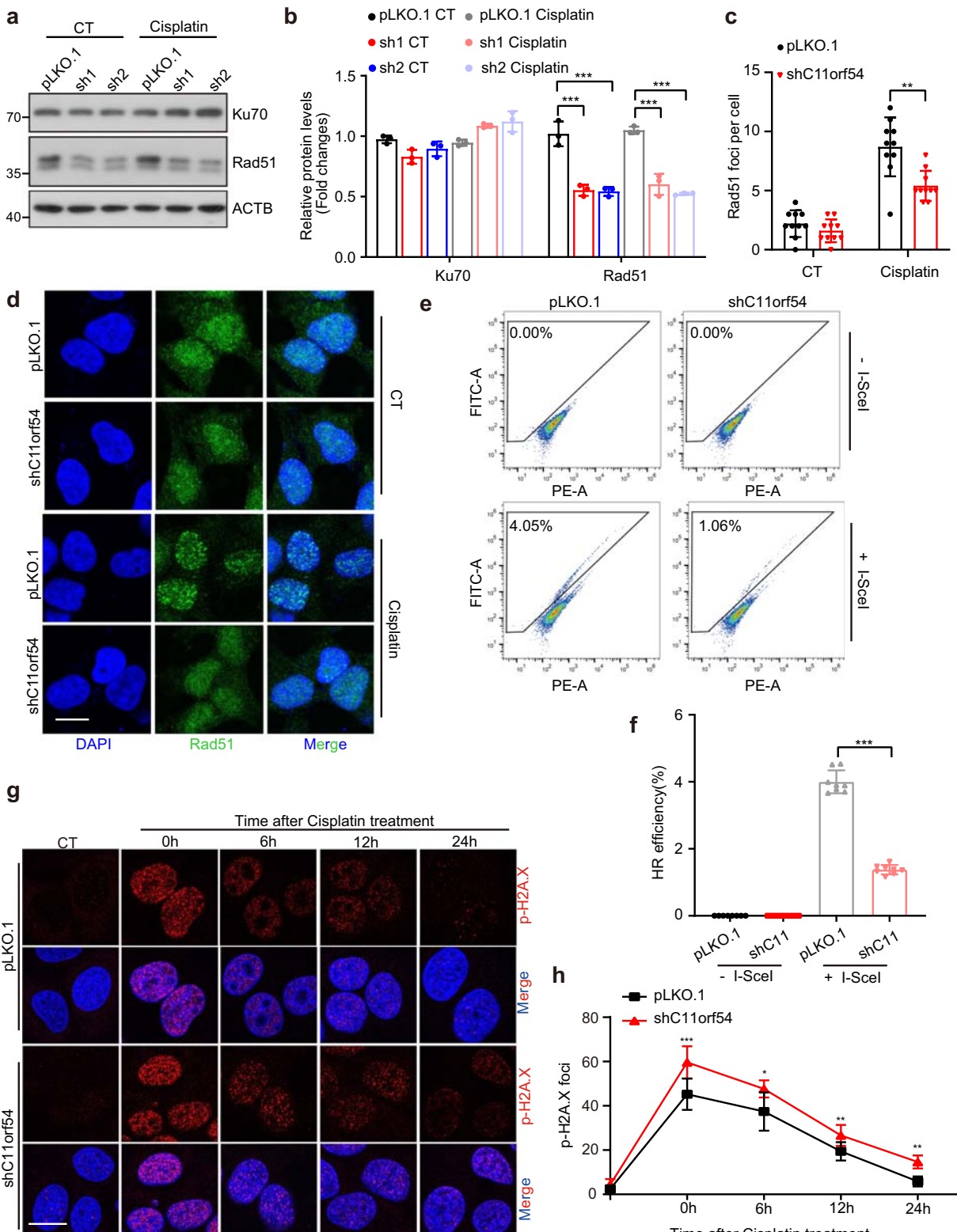

**Fig. 4 C11orf54 knockdown inhibits homologous recombination. a**, **b** Western blots (**a**) and quantitative results (**b**) of the indicated proteins in control and C11orf54 knockdown PLC/PRF/5 cells upon 10 μM cisplatin treatment for 6 h (*n* = 3 biological replicates; data are presented as mean values ± SD; ***p < 0.001 ). **c**, **d** Representative images of Rad51 foci (**d**) and quantitative results (**c**) in control and C11orf54 knockdown cells at the indicated time point after the cisplatin treatment (*n* = 10 independent cells; data are presented as mean values ± SD; **p < 0.01). Scale bar = 10 μm. **e**, **f** Flow cytometric analysis (**e**) and quantitative results (**f**) of the GFP-positive cells in control and C11orf54 knockdown PLC/PRF/5 cells after co-transfected pDR-GFP and pCBA-SceI (*n* = 3 biological replicates; ***p < 0.001). **g**, **h** Representative images of p-H2A.X foci (**g**) and quantitative results (**h**) in control and C11orf54 knockdown PLC/PRF/5 cells at the indicated time point after the cisplatin treatment (*n* = 10 independent cells; data are presented as mean values ± SD; *p < 0.05, **p < 0.01, ***p < 0.001). Scale bar = 10 μm.

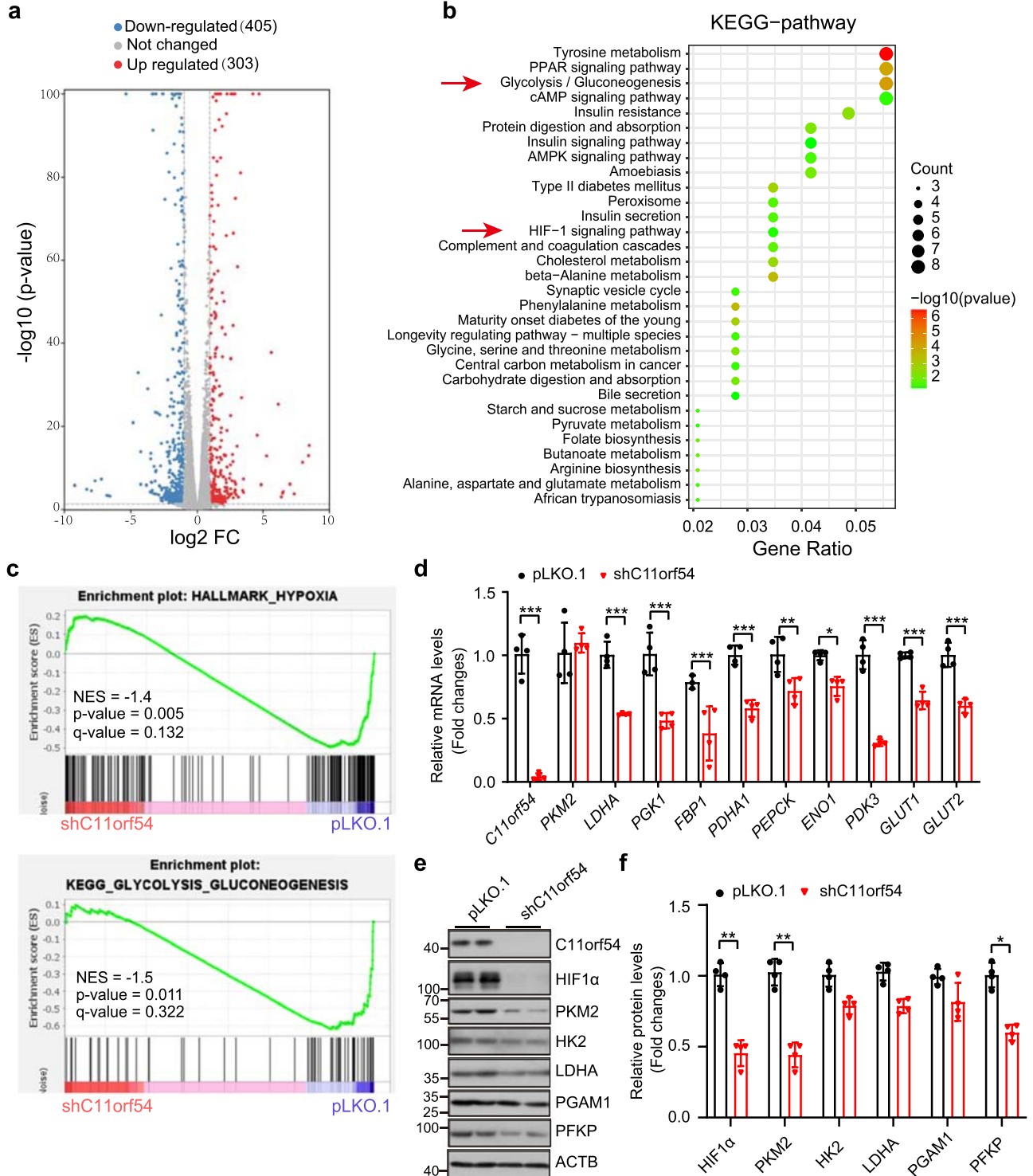

**Fig. 5 C11orf54 knockdown suppresses HIF1A and glycolysis signaling pathway. a** Volcano plot showing the significantly changed genes in C11orf54 knockdown versus control sample (red dot: upregulated genes, blue dot: downregulated genes, gray dot: no significant changed genes). **b** The top 20 functionally enriched KEGG pathways found in the analysis of DEGs in C11orf54 knockdown versus control sample. **c** Gene set enrichment analysis (GSEA) show hypoxia and glycolysis-gluconeogenesis signaling pathway has a trend to enrich the C11orf54 knockdown group. **d** qPCR experiment analysis of the mRNA expression of the glycolysis genes in C11orf54 knockdown and control cells ($n = 4$ biological replicates, data are presented as mean values ± SD, $*p < 0.05$, $**p < 0.01$, $***p < 0.001$). **e, f** Western blots (**e**) and quantitative results (**f**) of the glycolysis proteins in C11orf54 knockdown and control cells ($n = 4$ biological replicates; data are presented as mean values ± SD, $*p < 0.05$, $**p < 0.01$).

C11orf54 regulates HIF1A expression. Since C11orf54 knockdown does not affect the mRNA level of HIF1 (Fig. 6a), we evaluated the degradation of HIF1A. First, we treated the cell with the PHD inhibitor CoCl₂, which resulted in HIF1A stabilization and accumulation. However, C11orf54 knockdown consistently decreased the HIF1A expression upon CoCl₂ treatment (Supplementary Fig. 5a), suggesting that knockdown of C11orf54 repressed HIF1A may not be via the regulation of HIF1A

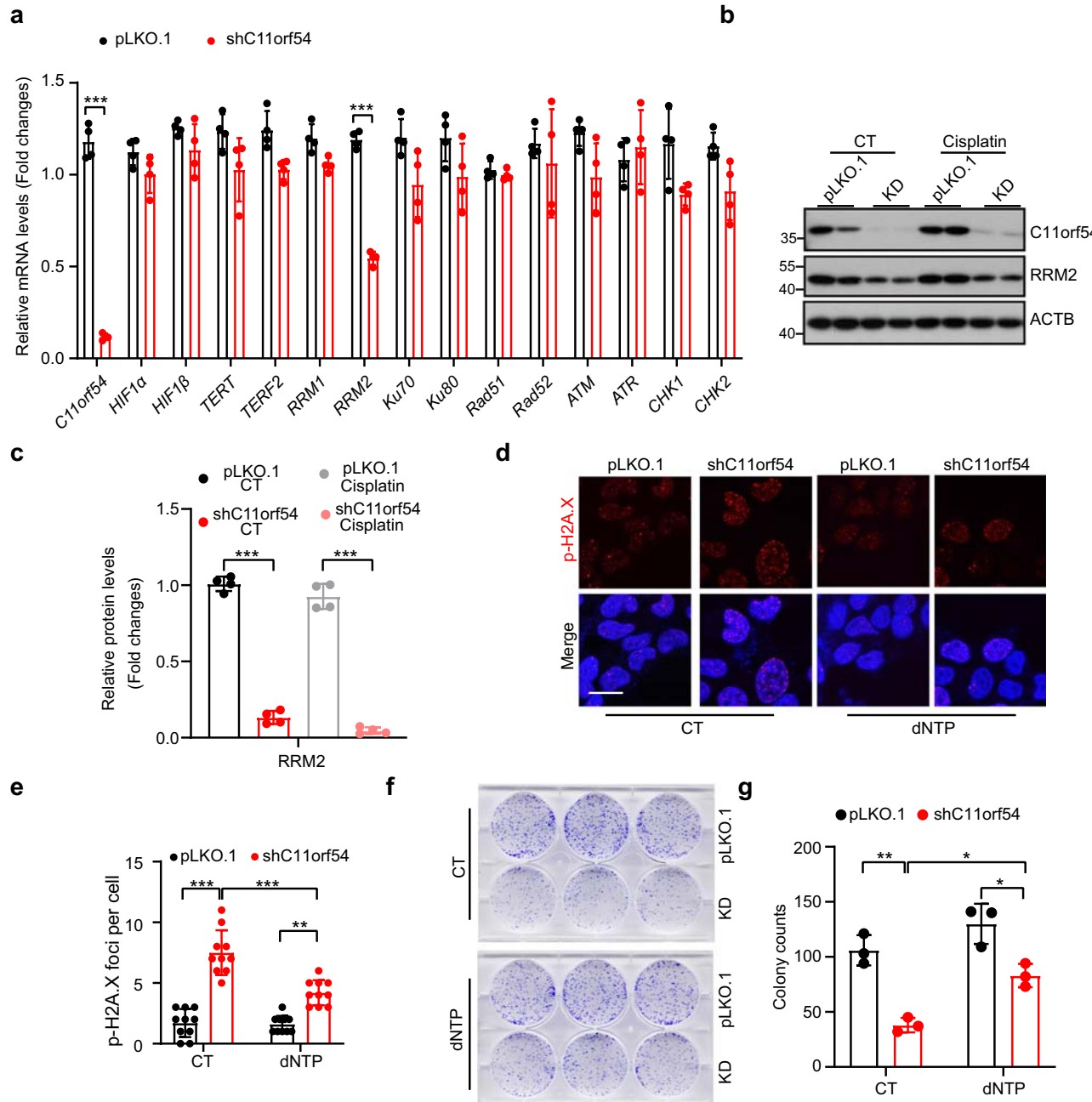

**Fig. 6 C11orf54 knockdown represses RRM2 expression. a** qPCR analysis of the mRNA expression of HIF1A downstream DNA repair-target genes in C11orf54 knockdown and control cells ($n = 4$ biological replicates, data are presented as mean values ± SD, ***$p < 0.001$). **b, c** Western blots (**b**) and quantitative results (**c**) of the indicated proteins in control and C11orf54 knockdown cells upon 10 μM cisplatin treatment for 6 h ($n = 4$ biological replicates; data are presented as mean values ± SD, ***$p < 0.001$). **d, e** Representative images of γH2A.X foci (**d**) and quantitative results (**e**) in control and C11orf54 knockdown cells upon 100 μM dNTP treatment for 24 h. ($n = 10$ independent cells; data are presented as mean values ± SD; **$p < 0.01$, ***$p < 0.001$). Scale bar = 10 μm. **f, g** Colony formation (**f**) and quantitative results (**g**) show the cell growth of C11orf54 knockdown PLC/PRF/5 cell and control cells under 100 μM dNTP treatment for 24 h ($n = 3$ biological replicates; data are presented as mean values ± SD, *$p < 0.05$, **$p < 0.01$).

stabilization. Then we wondered whether knockdown of C11orf54 causes HIF1A degradation through the proteasome or autophagy-lysosome pathway by using the MG132 and BafA1. We found that HIF1A expression was restored after BafA1 treatment but not MG132 (Fig. 7a, b). BafA1 is a widely used inhibitor for autophagosome-lysosome fusion and autolysosome acidification, suggesting that C11orf54 might promote HIF1A degradation through an autophagy-dependent manner.

Indeed, C11orf54 knockdown significantly increased the ratio of LC3B-II/I and decreased the autophagy substrate p62 upon

starvation and rapamycin treatment (Fig. 7c, d). Additionally, the number of autophagosome GFP-LC3 puncta increased in C11orf54 knockdown cells under normal and BafA1 treatment conditions (Fig. 7e, f). Furthermore, pretreatment with BafA1 could restore the cell viability and colony formation ability in C11orf54 knockdown cells (Fig. 7g–i). However, the treatment of PI3K inhibitors (Wortmannin and 3-MA), which repressed the early phase of autophagy, could not restore the protein level of HIF1A in C11orf54 knockdown cells (Supplementary Fig. 5b, c). On the contrary, treatment with a lysosomal inhibitor ($NH_4Cl$)

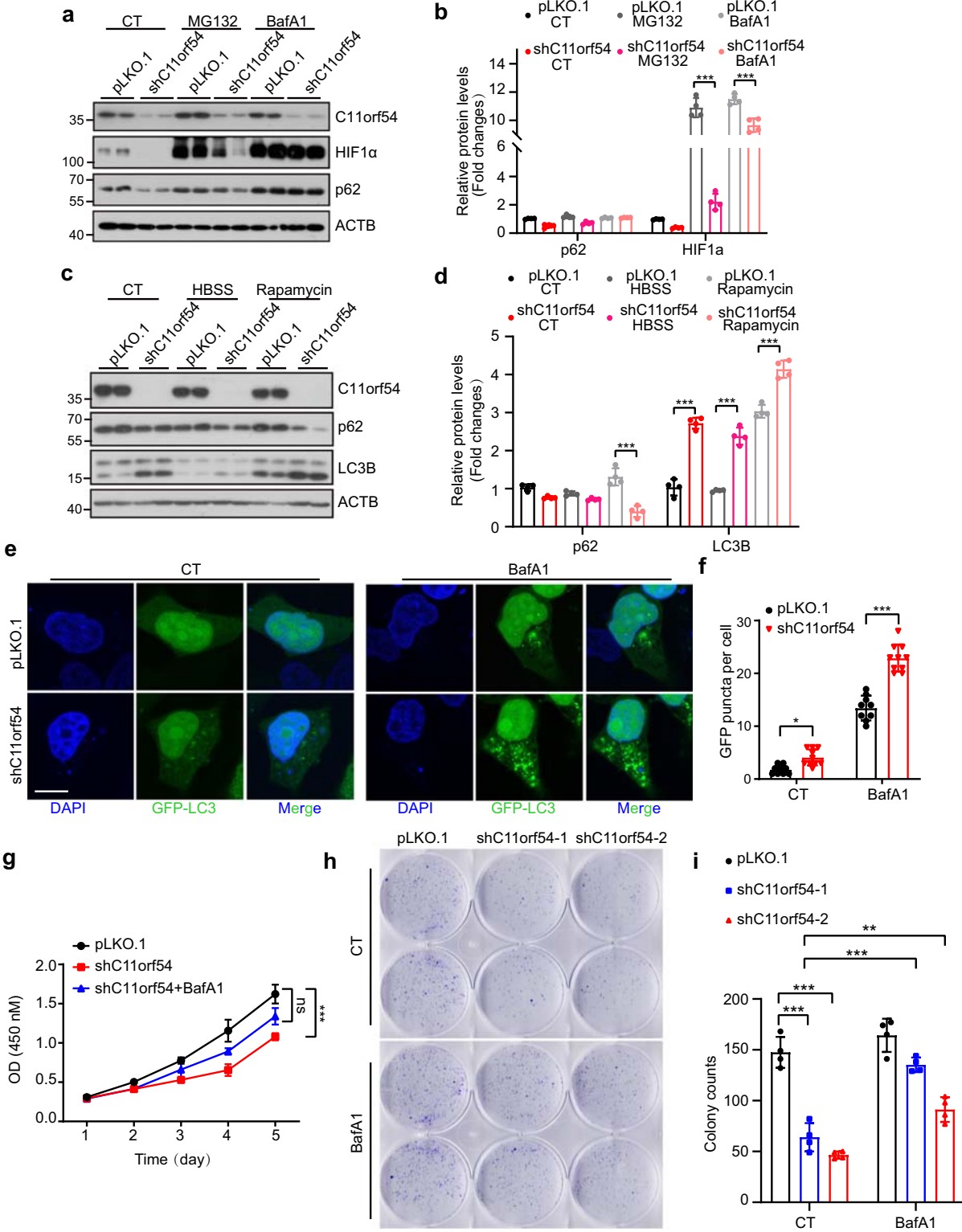

**Fig. 7 C11orf54 knockdown causes autophagy. a, b** Western blots (**a**) and quantitative results (**b**) of the indicated proteins in control and C11orf54 knockdown cells upon 10 μM MG132 and 100 nM BafA1 treatment for 8 h ($n = 4$ biological replicates; data are presented as mean values ± SD, \*\*\*$p < 0.001$). **c, d** Western blots (**c**) and quantitative results (**d**) of the indicated proteins in control and C11orf54 knockdown cells upon HBSS and 1 μM Rapamycin treatment for 12 h ($n = 4$ biological replicates; data are presented as mean values ± SD, \*\*\*$p < 0.001$). **e, f** Representative images (**e**) and quantitative results (**f**) of GFP-LC3 puncta in control and C11orf54 knockdown cells after 48 h transfection under normal and 100 nM BafA1 treatment for 8 h. (Scale bar = 20 μm; data are presented as mean values ± SD, \*$p < 0.05$, \*\*\*$p < 0.001$). **g** CCK8 assay shows the cell survival of C11orf54 knockdown PLC/PRF/5 cell and control cells upon normal and 0.5 nM BafA1 pretreatment for 12 h (data are presented as mean values ± SD, \*\*\*$p < 0.001$). **h, i** Colony formation (**h**) and quantitative results (**i**) show the cell growth of C11orf54 knockdown PLC/PRF/5 cell and control cells upon normal and 0.5 nM BafA1 pretreatment for 12 h ($n = 4$ biological replicates; data are presented as mean values ± SD, \*\*$p < 0.05$, \*\*\*$p < 0.001$).

could rescue the expression of HIF1A (Supplementary Fig. 5d). These results suggest that knockdown of C11orf54 reduces HIF1A expression through late phase of autophagy and lysosomes.

A recent study showed that HIF1A contained a KFERQ-like motif, which could be recognized and bound by HSC70, and then targeted to the LAMP2A multimeric complex on lysosome for chaperone-mediated autophagy (CMA) degradation[45]. Thus, we wonder whether C11orf54 regulates HIF1A degradation via chaperone-mediated autophagy. Firstly, we detected the expression of the core components of the CMA machinery (HSC70 and LAMP2A). However, C11orf54 does not affect the mRNA and protein levels of HSC70 and LAMP2A (Supplementary Fig. 6a–d).

Then we speculated that C11orf54 might influence the HIF1A targeting to lysosome and subsequently CMA-mediated degradation. Thus, we performed a co-immunoprecipitation/mass spectrometry (Co-IP/MS) assay to identify the proteins that potentially interact with C11orf54 (Supplementary Fig. 6e). The top identified C11orf54 interaction protein list was uploaded to the STRING online database to construct the protein-protein interaction (PPI) network. Then the hub genes selected from the PPI network using the maximal clique centrality (MCC) algorithm and cytoHubba plugin by the Cytoscape software are shown in Fig. 8a. The top 5 hubs genes were HSPA8, HSPA5, HSP90AA1, HSP90AB1 and HSPA9 (Fig. 8a, b). HSPA8 and HSP90AA1 both localize at the lysosomal membrane, from where they modulate different steps of CMA[46]. HSPA8 (HSC70) is responsible for substrate targeting for CMA[46]. The endogenous co-immunoprecipitation assay confirmed that C11orf54 interacted with HSC70 but not LAMP2 (Fig. 8c). In addition, by the KFERQ finder V0.8 online software, we recognized that C11orf54 contains two KFERQ-like motifs, which belong to the phosphorylation-activated motif and acetylation-activated motif, respectively (Supplementary Table 3). HIF1A contain a KFERQ-like motif, which could be recognized and interacted by HSC70. Thus, we hypothesized that C11orf54 and HIF1A competitively interact with HSC70. The interaction between HIF1A and HSC70 was restrained in the presence of C11orf54 (Fig. 8d). Meanwhile, the binding of HIF1A and HSC70 was weakened with the C11orf54 expression increasing (Fig. 8e). Furthermore, the interaction between HIF1A and HSC70 was reduced in the C11orf54 knockdown cells, possibly due to the decreased HIF1A expression. When treated with BafA1, the expression of HIF1A and its interaction with HSC70 were rescued (Fig. 8f). Furthermore, the expression of HIF1A was restored by double knockdown of C11orf54 and LAMP2A (Fig. 8g). Together, these data demonstrate that knockdown of C11orf54 promotes CMA-mediated HIF1A degradation by enhancing the interaction between HIF1A and HSC70.

## Discussion

In this study, we aimed to delineate a novel biological effect of C11orf54 in mammals. We demonstrated that C11orf54 knockdown decreased cell proliferation and enhanced cisplatin-induced DNA damage and apoptosis. Mechanistically, C11orf54 and HIF1A competitively interact with HSC70, the critical effector of chaperone-mediated autophagy, and knockdown of C11orf54 promotes CMA-mediated degradation of HIF1A. Moreover, C11orf54 knockdown-mediated HIF1A degradation reduced the transcription of ribonucleotide reductase regulatory subunit M2 (RRM2), which is a rate-limiting RNR enzyme for DNA synthesis and DNA repair by producing dNTPs. Supplement of dNTPs could partially rescue C11orf54 knockdown-mediated DNA damage and cell death. Furthermore, we found that autophagic inhibitor BafA1 could rescue the protein level of HIF1A and

mRNA level of RRM2, then restore the reduced cell proliferation of C11orf54 knockdown cells. On the other hand, loss of C11orf54 reduced Rad51 expression and nuclear accumulation, which resulted in suppression of homologous recombination repair (Fig. 9).

C11orf54 is a highly conservative gene in different species and is abundant in kidney and liver tissues. The biological role of C11orf54 was unclear, and the subcellular localization was still controversial. Conrad et al. revealed its nuclear location by an automatic phenotyping approach[27]. However, C11orf54 was predominantly localized in the cytoplasm, and when transfected with C11orf54-eGFP, it could exist in both cytoplasm and nucleus[28]. Here, we tested the localization of endogenous C11orf54 with two verified antibodies by the nuclear/cytosolic-fractionation assay and immunostaining analysis. The results indicated that C11orf54 was mainly located in the cytoplasm (Fig. 1d–g). C11orf54 was identified as a biomarker protein of endometrial cancer, renal cell carcinoma, and clear cell renal cell carcinoma by two-dimensional gel electrophoresis coupled with mass spectrometry[24–26]. However, C11orf54 was downregulated in renal cell carcinoma tissues compared to the corresponding normal tissues[24–26]. In addition, by analyzing the expression of C11orf54 in several cancer tissue samples based on TCGA data using the GEPIA website, we found that C11orf54 had low expression in most cancer tissues, especially in KICH (Kidney Chromophobe), KIRP (Kidney renal papillary cell carcinoma), and SARC (Sarcoma; Supplementary Fig. 7). In our study, C11orf54 knockdown could cause DNA damage and inhibit proliferation (Figs. 2–3). These suggest that the c11orf54 expression is decreased in cancer tissue through an unknown mechanism, which may be a feedback loop to block the tumor cell survival.

Homologous recombination (HR) and non-homologous end-joining (NHEJ) are the major pathways for DSB repair; Rad51 and Ku70 are two key regulators, respectively[47,48]. In our study, C11orf54 knockdown inhibits the expression of Rad51 rather than ku70, which implies that C11orf54 may influence HR repair. We used the HR reporter system and found that C11orf54 knockdown inhibited homologous recombination ability. Meanwhile, the clearance of p-H2A.X foci with the time course of recovery was inhibited after the C11orf54 knockdown.

Proteomic studies have shown that C11orf54 homolog protein increased in insulin resistance mice skeletal muscle, suggesting it may relate to protein folding/degradation signaling pathway[49]. Our results demonstrate that C11orf54 deficiency promotes the degradation of macroautophagy substrate p62 and the accumulation of LC3-II, which means the activation of macroautophagy (Fig. 6c–f). The autophagy-lysosome pathway is one of the two main routes for intracellular protein degradation[50].

In addition, we found that C11orf54 knockdown promoted CMA-mediated HIF1A degradation (Fig. 7c–f). Mechanistically, C11orf54 could competitively interact with HSC70 to dissociate HIF1A-HSC70 interaction. Knockdown of C11orf54 abolished its interaction with HSC70, resulting in enhanced interaction between HSC70 and HIF1A, eventually promoting CMA medicated HIF1A degradation (Fig. 8e, f). Consistently, a previous study showed that HIF1A could be degraded by CMA via interacting with HSC70 and LAMP2A[45]. Previous studies showed that macroautophagy and chaperone-mediated autophagy are complementary in protein degradation[51]. Recent studies suggested that when cells encounter stressful stimuli, macroautophagy and CMA could be activated[52,53]. Here, we proved that loss of C11orf54 could activate both macroautophagy and chaperone-mediated autophagy, but the mechanism of how C11orf54 regulated autophagy was still unclear, which needs further study in future work.

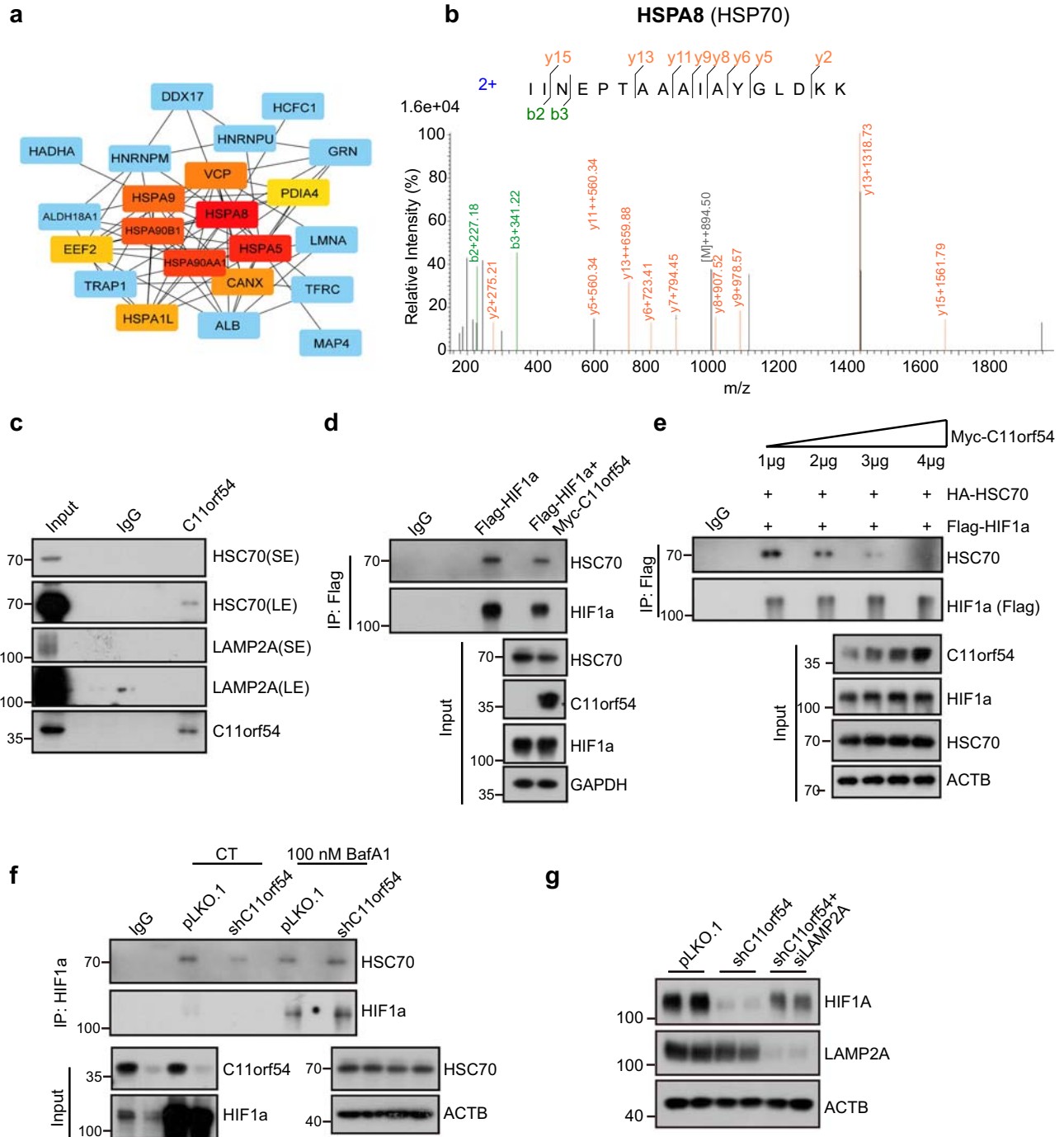

**Fig. 8 C11orf54 knockdown promotes CMA-mediate HIF1A degradation by enhancing the interaction between HIF1A and HSC70. a** The protein-protein interaction network analysis of C11orf54 binding proteins identified by Co-IP MS. **b** Mass spectrum of HSPA8 image. **c** Immunoprecipitation /Immunoblot analysis of the whole-cell lysates (WCLs) and Anti-C11orf54 derived from PLC/PRF/5 cells. IgG served as a negative control. **d** Immunoprecipitation/ Immunoblot analysis of the whole-cell lysates (WCLs) and Anti-Flag derived from the 293T cells transfected with Flag-HIF1A and Flag-HIF1A plus pk-Myc-C11orf54 for 48 h. IgG served as a negative control ($n = 3$ biological replicates). **e** Immunoprecipitation /Immunoblot analysis of the whole-cell lysates (WCLs) and Anti-Flag derived from the 293T cells transfected with Flag-HIF1A, HA-HSC70 and gradient pk-Myc-C11orf54 for 48 h. IgG served as a negative control ($n = 3$ biological replicates). **f** Immunoprecipitation/Immunoblot analysis of the whole-cell lysates (WCLs) and anti-HIF1A derived from the PLC/PRF/5 cells under control and 100 nM BafA1 treatment for 8 h. IgG served as a negative control ($n = 3$ biological replicates). **g** Western blots of the indicated proteins in control, C11orf54 knockdown and C11orf54, LAMP2A double knockdown cells.

A previous study revealed that the activation of HIF1A/ STAT3 signaling could upregulate RRM2 mRNA expression[44], and we also discovered that inhibition of HIF1A degradation with BafA1 could partially rescue RRM2 at the mRNA level. RRM2 level fluctuates during the cell cycle, which is increased during the late G1/early S phase and is degraded in the late S phase. RRM2 levels are kept in check by the APC$^{Cdh1}$ ubiquitin ligase to prevent RRM2 accumulation in G1 phase[54]. Additionally, in G2 phase, following CDK-mediated phosphorylation, RRM2 is degraded via Cylin F[55]. Furthermore, a recent study showed that

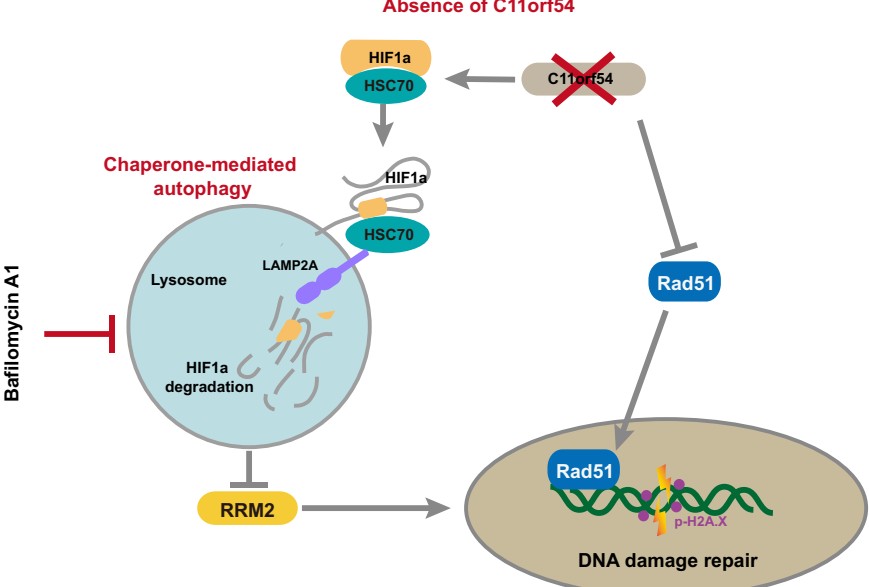

**Fig. 9 Schematic model of how C11orf54 promotes DNA repair via blocking CMA-mediated degradation of HIF1A.** On the one hand, loss of C11orf54 reduces Rad51 expression and nuclear accumulation. On the other hand, knockdown of C11orf54 promotes HSC70 binding to HIF1A to target it for degradation via chaperone-mediated autophagy (CMA), which results in downregulation of RRM2. Eventually, absence of C11orf54 causes suppression of homologous recombination repair.

downregulation of RRM2 by siRNA or treatment with the RNR inhibitor hydroxyurea substantially induced autophagy. However, RRM2 is targeted for proteasome-dependent but autolysosome-independent degradation upon induction of autophagy[56]. Our results showed that C11orf54 knockdown reduced RRM2 level through decreased mRNA and protein level (Fig. 6a–c). Supplement of dNTPs could not rescue Rad51 expression but could partially recover C11orf54 knockdown-mediated DNA damage and cell proliferation, which means there might be an alternative mechanism effecting HR. Meanwhile, knockdown of C11orf54 promotes autophagy (Fig. 7). Hydroxyurea is a well-established inhibitor of RNR by reducing the free radical and the iron center of RRM2[57]. In our study, Hydroxyurea, Camptothecin and Cisplatin promote the expression of C11orf54 in a time-dependent manner (Fig. 2f, g and Supplementary Fig. 3a–d). Thus, what's the exact regulation network among C11orf54, RRM2, and autophagy also need further study.

In summary, we uncoverd a role of C11orf54 in regulating DNA damage and repair through CMA-mediated degradation of HIF1A, resulting in reduced RRM2. C11orf54 could competitively interact with HSC70 and suppress the HIF1A target to CMA degradation. BafA1 or dNTPs treatment could partially rescue C11orf54 knockdown-mediated DNA damage and proliferation inhibition.

## Methods

**Cell culture**. PLC/PRF/5 and 293T cells were purchased from Shanghai Cell Bank, Type Culture Collection Committee, Chinese Academy of Sciences. All these cells were grown in Dulbecco's modified Eagle's medium (DMEM) supplemented with 10% fetal bovine serum (Hyclone) and 1% penicillin-streptomycin(Gibco) at 37 °C under 5% $CO_2$.

**Plasmids**. For overexpression plasmid, the C11orf54 CDS was cloned from 293T cell cDNAs using the PCR amplification primers listed in Supplementary Table 1. The pk-Myc vector was digested with BamHI and NotI to linearize the vector and then ligated C11orf54 CDS and linearized vector with T4 ligase.

For shRNA design, using BLOCK-iT™RNAi Designer (http://rnaidesigner. thermofisher.com/) to determine two top-scoring targets for C11orf54. Two target sequences are as follows:

shC11orf54-1: 5′-GGTGCCTACTGGAGAAATACA-3′;
shC11orf54-2: 5′-CCAGGTCTCTGTAGTTGATTG-3′.

The pLKO.1 vector was digested with EcoRI and AgeI, and then ligated with shRNA oligos by the T4 ligase.

**Lentivirus package and transfection**. To prepare lentivirus, we co-transfected shC11orf54 plasmids into 293T cells with psPAX2 and pMD2.G, using poly-ethylenimine (PEI) transfection reagent. The lentiviral supernatants were harvested at 48 and 72 h after transfection, then infected PLC/PRF/5 and 293T cells. The cells were selected in 1 μg/ml puromycin-containing medium 3 days after infection.

**Quantitative real-time PCR**. Total RNA was extracted from cells using the RNAiso Plus regent (TaKaRa) following the manufacturer's instructions. RNA was reverse transcribed into cDNA using the ABScript II First Strand Synthesis Kit (Abclonal) following the manufacturer's instructions. Quantitative reverse transcription PCR (qRT-PCR) was carried out with SYBR Green Master Mix (Abclonal) on a CFX96 real-time system (Bio-Rad). Relative mRNA levels were calculated using the $2^{-\Delta\Delta Ct}$ method, with Actb used as the internal control. Primer sequences for target genes are shown in Supplementary Table 1.

**Western blotting**. Proteins were isolated in ice-cold RIPA buffer (Beyotime) with proteinase inhibitors, and protein concentrations were determined by the bicinchoninic acid assay (BCA). Proteins were fractionated by sodium dodecyl sulfate-polyacrylamide gel electrophoresis, electroblotted onto the poly- (vinylidene difluoride) membrane (Millipore), and probed with primary and secondary antibodies. The primary and secondary antibodies used are shown in Supplementary Table 2. The protein bands detected by the antibodies were visualized by enhanced chemiluminescence (Beyotime) and evaluated using Image J.

**Immunofluorescence staining**. Cells were grown to 60% confluence on a cover-slip. After treatment, cells were fixed with 4% paraformaldehyde at room temperature for 20 min. Antigen accessibility was increased by treatment with 0.2% Triton X-100 and blocked with 3% bovine serum albumin. Cells were incubated with primary antibodies overnight at 4 °C. After washing with PBST, stained with Alexa Fluor 594-conjugated anti-rabbit IgG for 1 h in the dark at room temperature. After DAPI staining, the cells were imaged with a Leica TCS SP8 confocal microscope. For colocalization quantification, images were preprocessed by subtracting a median filter-processed image and background, and then images proceeded with Image J.

**Nuclear/cytosolic-fractionation assay**. The Nuclear/cytosolic-fractionation assays were performed using Nuclear and Cytoplasmic Protein Extraction kit (Beyotime) following the manufacturer's instructions. Briefly, $2 \times 10^6$ cells were vortexed with cytoplasm extraction Reagent A for 5 s and incubated on ice for 15 min. Adding cytoplasm extraction Reagent B and incubated on ice for 1 min after 5 s vortex, then centrifuged at 4 °C for 5 min at 12,000 g. The supernatant was

cytosolic fractionation. Then the nuclear pellet was washed three times with iced-PBS and resuspended with nuclear extraction reagent and then vortexed every 2 min for 30 s for a total 30 min. Centrifuging at 4 °C for 10 min at 12,000 g, and the supernatant was nuclear fractionation.

**Co-immunoprecipitation assays.** PLC/PRF/5 and 293T cells were seeded in 10-cm dishes and transfected with the indicated plasmids. Two days after transfection, cells were lysed with Western/IP buffer on ice and then sonicated. Protein concentrations were determined in the bicinchoninic acid assay (BCA). Flag or HIF1α antibodies were used to immunoprecipitate endogenous HSC70. The precipitates were boiled and loaded onto SDS-PAGE gels for western blot with secondary antibody VeriBlot for IP detection.

**Annexin V(FITC)/propidium iodide double staining apoptosis assay.** Cell apoptosis was assessed by flow cytometry using the Annexin V-FITC Apoptosis Detection Kit (Beyotime) following the manufacturer's instructions. Briefly, cells after treatment were trypsinized, washed with PBS, resuspended in binding buffer, and incubated with staining solution(annexin V/PI = 2:1)in the dark for 20 min at room temperature. Immediately after the annexin V/PI staining, fluorescence-activated cell sorting (FACS) analysis was performed using BD FACS VERSE.

**TUNEL staining.** The TUNEL staining was using One Step TUNEL Apoptosis Assay Kit (Beyotime) following the manufacturer's instructions. Briefly, after 24 hr Cisplatin treatment, cells were fixed with 4% paraformaldehyde for 20 min and then permeabilized with 0.2% Triton X-100 for 5 min at room temperature. After washing with PBS, incubating cells with TUNEL test solution (labeling buffer/TdT enzyme = 9:1) in the dark for 60 minutes at 37°C. Finally, cells were imaged with Zeiss Axio Imager.Z2.

**CCK8 assay.** CCK-8 (Solarbio) was used to assess the cell proliferation following the manufacturer's instructions. Briefly, cells were seeded in 96-well plates at the density of $1 \times 10^3$ cells/well, then cultured in an incubator for 24 h before being evaluated at day-1, -2, -3, -4, -5, respectively. CCK-8 solution was then dripped into each well, and the plate was transferred to the incubator for two hours. Finally, an OD value at 450 nm was detected by MD SpectraMax 190.

**Colony formation assay.** About $1 \times 10^3$ pLKO.1 and shC11orf54 PLC/PRF/5 cells were seeded in a six-well plate and treated with the indicated concentration of cisplatin. The cells were cultured for 10 days, and 4% paraformaldehyde was used to fix the cells, followed by staining with 0.5% crystal violet for 1 hour. The number of colonies ( > 50cells/colony) was counted using a stereomicroscope and analyzed by image J software[58]. All the samples were done in triplicate.

**EdU staining.** The EdU staining was using BeyoClick™ EdU-594 Kit (Beyotime) following the manufacturer's instructions. Briefly, 10 μM EdU was added into the fresh medium and then incubated for 2 hours at 37 °C/5% $CO_2$ when cells were grown to 80% confluence. After incubation, the cells were washed with PBS and fixed with 4% paraformaldehyde at room temperature for 20 minutes. Then the cells were incubated with reaction buffer in the dark for 30 min at room temperature. After DAPI staining, the cells were imaged with Nikon Ti2 to visualize the number of EdU-positive cells. The positive rate was determined by Image Pro Plus 6.

**Comet assay.** The Comet assays were performed as in our previous study using COMET Assay kit (Enzo) following the manufacturer's instructions[59]. Combine cells at $1 \times 10^5$/ml with molten LM agarose at a ratio of 1:10 (vol/vol) and immediately pipetted onto a COMET slide. Placed the slides flat at 4 °C in the dark for 30 min and then immersed in the pre-chilled lysis solution at 4 °C for 30 min. Removed the slides, gently taped excess buffer from slides, washed in TBE buffer, and then transferred to a horizontal electrophoresis chamber. Voltage (1 V/cm) was applied for 10 min. Very gently taped off excess TBE, dipped slides in 70% ethanol for 5 minutes, and air dry samples. Slides were stained with SYBR Green and then analyzed by fluorescence microscopy. In all, 70–150 cells were evaluated in each sample using the COMET Assay Software Project (CASP software). Tail DNA% = Tail DNA/ (Tail DNA + Head DNA), Tail moment = Tail length × Tail DNA%.

**Measurement of homologous recombination repair.** The HR repair activity assay was performed as the previous study[37]. Briefly, pLKO.1 and shC11orf54 PLC/PRF/5 cells were transfected with pDR-GFP and pCBA-SceI. After two days, cells were harvested and analyzed by fluorescence-activated flow cytometry (FACS) to examine the percentage of GFP-positive cells. The gating strategies are shown in Supplementary Fig.8.

**RNA-seq analysis and gene set enrichment analysis.** Total RNA was extracted from the pLKO.1 and shC11orf54 PLC/PRF/5 cells. The RNA was then sequenced by the WuXiNextCODE Tec RNA-seq service ($n = 2$). GO analysis of the Differentially Expressed Genes significant changes was performed using Metascape website (https://metascape.org/gp/index.html#/main/step1)[60]. For gene set enrichment analysis, we applied GSEA v4.1.0 to various functional characteristic gene signatures as described previously[61,62]. GSEA was performed using the "Hallmark" or "KEGG" gene sets to identify enriched signatures. Gene Sets with an FDR < 0.25 and a nominal $p$-value of <0.05 were considered significant.

**Statistics and reproducibility.** The results are expressed as the mean ± standard deviation (SD). The level of statistical significance was set at $p < 0.05$ using an unpaired two-tailed Student's $t$ test. *$p < 0.05$; **$p < 0.01$; ***$p < 0.001$. All statistical analyses were performed using GraphPad Prism software. The sample and replicate size were indicated in the figure legends.

**Reporting summary.** Further information on research design is available in the Nature Portfolio Reporting Summary linked to this article.

## Data availability

The RNA-seq data generated in this study have been deposited in the Sequence Read Archive (SRA) database under accession code PRJNA939822, sample IDs: SRR23648017, SRR23648018, SRR23648019 and SRR23648020. The unedited/uncropped western blot gels are included in Supplementary Fig. 9. The source data behind the graphs in the paper are included in Supplementary Data 1. All other data are available from the corresponding authors on reasonable request.

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

## Acknowledgements

We thank Dr. Guo Chen from Jinan University for the pDR-GFP and pCBA-SceI plasmids. This work was financially supported by the National Key R&D Program of China (2021YFA0804903), the National Natural Science Foundation of China (Grants. 32170772, 32270810, 81800833, and 81802189), the 111 Project (B16021), and the Natural Science Foundation of Guangdong Province (grant 2021A1515011227, 2022A1515140040, 2019A1515011847, and 2019A1515010591). Q.Z. also gratefully acknowledges the support of K.C. Wong Education Foundation.

## Author contributions

The project was supervised by J.L. and Q.Z. Experiments were designed by J.L. and Q.Z., performed by J.T., W.W., X.L., J.X., Y.C., Y.L., J.H., and L.H.; J.T., W.W., and J.L. analyzed the data. J.T., W.W., J.L., and Q.Z. wrote the manuscript. All the authors reviewed the manuscript.

## Competing interests

The authors declare no competing interests.
