## [Peer Review File · Communications Biology]

Reviewers' comments:

Reviewer #1 (Remarks to the Author):

In the present work, Tan et al. have characterized the effects of C11orf54 down regulation on DNA damage/repair, cell death and proliferation, as well as the connection with HIF1A protein abundance. The authors find that C11orf54 down regulation induces DNA instability, cell proliferation arrest and apoptosis. The effect of C11orf54 on the cell phenotype mentioned before were boost by the incubation with cisplatin, a well-known anticancer drug that causes DNA damage and apoptosis. In general, the work is well written, and the applied techniques and results obtained are in accordance with a very good research work. Beyond minor corrections to be done, the major concern is about the role of C11orf54 in blocking the degradation of HIF1A through CMA. Also, more experiments are needed to confirm that HIF1A regulates RRM2 expression/protein abundance.

Minor corrections:

- 1- Line 65 is written in past, please review
- 2- Line 69, please review the redaction of the paragraph.
- 3- Line 74, please review the redaction of the paragraph.
- 4- Line 78, please review the redaction of the paragraph.
- 5- Line 87, please review the redaction of the paragraph (prolonged persistence)
- 6- Line 87: If CMA inhibition provokes the up regulation of HIF1A protein levels, why CMA inhibition also causes DNA damage and impairs DNA repair (according to this sentence). This is discordant.
- 7- Line 95: If C11orf54 is a cancer biomarker, then, its protein levels are augmented in cancer cells. If this is true, then the degradation of HIF1A through CMA should be impaired in cancer cells because of the high levels of C11orf54. It does not fit with the authors' hypothesis. Please discuss it.
- 8- Line 132: The immunostaining... This phrase could be move up, after line 126.

Major concerns:

- 1- In figure 2, it would be good to perform an apoptosis assay to show if cells, apart from stop the proliferation, goes in cell death.
- 2- Line 158: it seems that the effect of C11orf54 on the caspase 3 activation is only in presence of cisplatin. Can you please mention it on the results section and discuss.
- 3- Line 251: The rescue of the phenotypes by the addition of dNTPs does not prove that HIF1A is regulating RRM2 protein levels in these cells. Additional experiments must be performed to confirm it.
- 4- Why C11orf54 will compete only for the binding of HIF1A and no other HSC70 binding proteins. There are other HSC70 binding proteins, substrates of CMA, that are being displaced by C11orf54?
- 5- C11orf54 seems to have high affinity for HSC70, at least higher than HIF1A (to avoid its binding to HSC70). However, the C11orf54 KFERQ motifs are no classic. These motifs should be mutated to confirm the competition for HSC70.
- 6- If C11orf54 has high affinity for HSC70. So, why C11orf54 is not degraded by CMA? Authors should test it.
- 7- Even the use of BafA1 and other lysosomal function inhibitors, authors should knock down Lamp2A to block CMA activity and rescue the HIF1A protein levels (decreased by the knocking down of C11orf54).
- 8- Finally, in my opinion, the blocking of HSC70-HIF1A interaction by C11orf54 is not a function of the C11orf54 protein. This is a biological effect and should be described in a different way than a "function".

Reviewer #2 (Remarks to the Author):

Summary

Tan and colleagues show that depletion of C11orf54, a biomarker protein found in renal cancers decreases cell viability and that this effect seems to be further potentiated by cisplatin treatment. The authors show that depletion of C11orf54 leads to loss of RAD51 expression which then leads to a HR defect. Mechanistically they demonstrate that C11orf54 could competitively interact with HSC70 to dissociate HIF1A-HSC70 interaction. When C11orf54 is inhibited this leads to an increased association of HSC70 with HIF1A and as a result promotes CMA mediated HIF1A degradation. Increased HIF1A degradation then leads to reduced RRM2 expression and dNTP imbalance, as RRM2 is a transcription target of HIF1A. The authors attribute the dNTP imbalance as the cause of homologous recombination repair deficiency and genome instability. The findings of this study are novel and will be of interest in the field.

Major points

The mechanism of how C11orf54 regulates RRM2 and autophagy seems interesting but feel that there are several different mechanisms operating in response to loss of C11orf54 in this study. I think that each specific claim (listed in the paper) is experimentally shown. However, connections (causal relationships) among each finding are not well established. Thus, these observations can be interpreted in many different ways, as has been mentioned by the authors. For example how RAD51 expression is regulated by C11orf54 via HIF1A/RRM2 suppression is unclear. Moreover, if HIF1A/RRM2 defects were the causative factors causing loss of viability in C11orf54 depleted cells then overexpressing these should reverse these phenotypes. In addition, the authors do not show that; 1) HIF1A or RRM2 depletion causes loss of viability in these cells and 2) double depletion of C11orf54/HIF1A or C11orf54/RRM2 should be epistatic in terms of the decrease in cell viability if the mechanism is via this axis. These experiments must be carried out if the authors want to reach to the conclusion that the mechanism is via HIF1A/RRM2. Moreover, like C11orf54 depletion, does HIF1A or RRM2 depletion also cause cisplatin sensitivity or is this a separate mechanism isolated to C11orf54 loss.

Authors suggest a model which they think most likely explains the mechanism of how C11orf54 promotes DNA repair, but I think it includes too many hypothesis, which haven't been fully proven.

Specific points

Figure 2. Authors use two cancer cell lines to investigate the role of C11orf54, one kidney and one liver cancer. To demonstrate that depletion of C11orf54 causes cancer cell inhibition the authors must demonstrate the effect in a larger panel of cell lines. In particular the authors should use more renal cancer cell lines since C11orf54 has been shown to be a biomarker in this disease setting. What is the effect in normal cells? The authors should investigate if there is a relationship between high C11orf54 expression in renal cancer cell lines vs low expression and cell inhibition in response to C11orf54 depletion. In addition, using TCGA or another publically available dataset show how C11orf54 is a biomarker in cancer? show a figure illustrating expression across tumour types, as supplementary data.

The significant decrease in EdU incorporation in the C11orf54 depleted cells, suggests significant disturbance in S-phase. The authors should look at replication dynamics using fibre analysis in C11orf54 depleted cells vs wild type cells. Its very likely that C11orf54 has a role in replication since the cells are sensitive to replication blocking agents. Is the localization of C11orf54 altered in response

to DNA damage – for example is it now localized to the nucleus given the previous reports?

Figure 3. The authors state that the DNA damage signalling is ATM dependent, have they tested ATR as well since the signalling was only partially restored in response to ATM inhibitor?

Figure 4. The authors must look at RAD51 foci and chromatin association of RAD51. Does loss of C11orf54 depletion effect RAD51 at the protein or mRNA level?

If dNTP depletion via a defect in HIF1A/RRM2 is the cause of the reduction in RAD51 expression/HR frequency, does dNTP supplementation rescue RAD51 levels back to wild type in C11orf54 depleted cells? Can you rescue cisplatin sensitivity by supplementing C11orf54 depleted cells with dNTP? In addition, have the authors tested PARPi in their cells since PARPi induce lesions directly engage HR for repair? This should be tested. Its possible that HR frequency is effected due to a significant disturbance in S-phase in C11orf54 depleted cells.

Figure 5 Show a table of differentially expressed genes in the cell lines shown in this figure. Did RAD51 score in the RNAseq data? What about RRM2?

Figure 6. The authors need to clearly justify how they selected to look at the DNA repair factors in Figure 6A. There are many more DNA repair genes why were these particular ones selected? Did they come up in the RNAseq data? Why did you not also assess RAD51 here?

Does RRM2 depletion cause the same level of cell inhibition/DNA damage as depleting C11orf54, is co-depletion of RRM2/C11orf54 epistatic in terms of cell inhibition? What about HIF1A?

Figure 7. Does BafA1 treatment in C11orf54 depleted cells rescue the cell viability observed in response to cisplatin treatment?

There are many textual errors found throughout the paper including in the schematic figure of the mechanism (Figure 9) – should say “Absence of C11orf54” not “Absensent of C11orf54” and “HIF1A degradation” not “HIF1A degeradation”. The text throughout the paper must be thoroughly checked by the authors.

Minor point

The absence of something cannot promote – rephrase so that it says causes rather than C11orf54 depletion promotes throughout the text

Rephrase line 179-180 about ATM and ATR

Please rephase line 233-236

Figure 7 instead of using promotes please use causes

Point-by-point response

We appreciate the reviewers for considering the strengths of our work and for their valuable advice and suggestions for improving this manuscript. We have tried our best to address these points by conducting new experiments and revising the manuscript. Below are our point-by-point responses (*blue italic type*) to the reviewers' comments.

=====

Reviewer #1 (Remarks to the Author):

In the present work, Tan et al. have characterized the effects of C11orf54 down regulation on DNA damage/repair, cell death and proliferation, as well as the connection with HIF1A protein abundance. The authors find that C11orf54 down regulation induces DNA instability, cell proliferation arrest and apoptosis. The effect of C11orf54 on the cell phenotype mentioned before were boost by the incubation with cisplatin, a well-known anticancer drug that causes DNA damage and apoptosis. In general, the work is well written, and the applied techniques and results obtained are in accordance with a very good research work. Beyond minor corrections to be done, the major concern is about the role of C11orf54 in blocking the degradation of HIF1A through CMA. Also, more experiments are needed to confirm that HIF1A regulates RRM2 expression/protein abundance.

We would like to thank reviewer #1 who approves the significance of our paper and points out those writing errors. As requested, we have corrected the errors and carefully revised the manuscript.

Minor corrections:

- 1-Line 65 is written in past, please review
- 2-Line 69, please review the redaction of the paragraph.
- 3-Line 74, please review the redaction of the paragraph.
- 4-Line 78, please review the redaction of the paragraph.
- 5-Line 87, please review the redaction of the paragraph (prolonged persistence)

As required, we have corrected the tense in the revised manuscript.

6-Line 87: If CMA inhibition provokes the up regulation of HIF1A protein levels, why CMA inhibition also causes DNA damage and impairs DNA repair (according to this sentence). This is discordant.

Park et al has proved that CMA is upregulated in response to different genotoxic insults to assure timely release of cell cycle arrest after DNA repair. regulated degradation of Chk1 by CMA in response to DNA damage is required to ensure cell cycle progression, and that failure to efficiently eliminate Chk1 by this pathway leads to persistent cell arrest, accumulation of DNA damage and alterations in nuclear proteostasis (PMID: 25880015).

Besides, CMA modulates the levels of proteins involved in the cell cycle, such as pyruvate kinase (PKM2), which stimulates MYC and cyclin D for cell cycle progression in G1 (PMID: 21700219). p21 can also be modulated by Rho family GTPase 3 (RND3), another CMA substrate that induces cell cycle arrest in G1(PMID: 26761524). MYC is also induced through the cancerous inhibitor of protein phosphatase 2A (CIP2A), which is modulated by CMA, and is inhibited by RDN3 (PMID: 28410006). MYC is required for positive cell cycle regulation and progression through the cyclin-CDK induction. Mouse double-minute 2 homolog (MDM2) is a CMA substrate that inhibits p53 and p73 for cell cycle progression(Mouse double-minute 2 homolog (MDM2) is a CMA substrate that inhibits p53 and p73 for cell cycle progression; p73 is also controlled by CMA and promotes cell cycle arrest by inducing p21.

In hypoxia situations, the hypoxia-inducible factor-1 subunit alpha (HIF-1 α) is induced by CDK1 and inhibits MDM2 to trigger cell cycle arrest, HIF-1 α can be inhibited by CDK2 for cell cycle progression (PMID: 20540933). p73 is also controlled by CMA and promotes cell cycle arrest by inducing p21(PMID: 32971884).

Thus, the downstream function of CMA inhibition dependent on the target protein and conditions.

7-Line 95: If C11orf54 is a cancer biomarker, then, its protein levels are augmented in cancer cells. If this is true, then the degradation of HIF1A though CMA should be impaired in cancer cells because of the high levels of C11orf54. It does not fit with the authors' hypothesis. Please discuss it.

Thanks for raising this discussion. In fact, C11orf54 was downregulated in renal cell carcinoma tissues compared to the corresponding normal tissues by 2-dimensional polyacrylamide gel electrophoresis (2D-PAGE) (PMID: 20464042; PMID: 27022288). In addition, by analyzing the expression of C11orf54 in several cancer tissue samples based on TCGA data using the GEPIA website, we found C11orf54 had low expression in most cancer tissues, especially in KICH, KIRP and SARC. In our research, C11orf54 knockdown could cause DNA damage and inhibit proliferation. Thus, we think that the c11orf54 expression is decreased in cancer tissue through an unknown mechanism, which is a feedback loop to block the tumor cell survival.

Fig.1 The expression of C11orf54 in several cancer tissue samples based on TCGA data from the GEPIA website

8-Line 132: The immunostaining... This phrase could be move up, after line 126.

Thanks for your suggestion. However, in our opinion, we first designed shRNA and tested the efficiency of C11orf54 knockdown at protein and mRNA levels. Then we checked the antibody specificity by western blot assay. Finally, we performed immunostaining and nuclear/cytosol fractionation assay to confirm the subcellular localization of C11orf54. We successively conduct these experiments and think this order is logical.

Major concerns:

1-In figure 2, it would be good to perform an apoptosis assay to show if cells, apart from stop the proliferation, goes in cell death.

As suggested, we conducted TUNEL staining assay, which showed that C11orf54 knockdown promote apoptosis after 24hr 20μM cisplatin treatment.

Fig.2 Representative images of TUNEL assay (A) and quantitative results (B) in pLKO.1 and C11orf54 knockdown cells after 48hr 20μM Cisplatin treatment.

2-Line 158: it seems that the effect of C11orf54 on the caspase 3 activation is only in presence of cisplatin. Can you please mention it on the results section and discuss.

As recommended, we have mentioned and discussed the result in the revised manuscript.

3-Line 251: The rescue of the phenotypes by the addition of dNTPs does not prove that HIF1A is regulating RRM2 protein levels in these cells. Additional experiments must be performed to confirm it.

As suggested, we reintroduced HIF1A in C11orf54 knockdown cell and found that HIFA could partially rescue the expression of RRM2 after C11orf54 knockdown.

Fig.3 Western blots of the indicated proteins after HIF1A overexpression

4-Why C11orf54 will compete only for the binding of HIF1A and no other HSC70 binding proteins. There are other HSC70 binding proteins, substrates of CMA, that are being displaced by C11orf54?

To elucidate this problem, we executed new co-IP assay. Result indicates that HSC70 binding proteins like CHK1 and HK2 could also be displaced after C11orf54 overexpression.

Fig.4 Immunoprecipitation /Immunoblot analysis of the whole-cell lysates (WCLs) and Anti-HA derived from the 293T cells transfected with pk-Myc plus HA-HSC70 and pk-Myc-C11orf54 plus HA-HSC70 for 48 h. IgG served as a negative control.

5-C11orf54 seems to have high affinity for HSC70, at least higher than HIF1A (to avoid its binding to HSC70). However, the C11orf54 KFERQ motifs are no classic. These motifs should be mutated to confirm the competition for HSC70.

By the KFERQ finder V0.8 online software, we recognize that C11orf54 contains two non-classical KFERQ-like motifs, which belong to the phosphorylation-activated motif and acetylation-activated motif, respectively As recommended, we constructed the two KFERQ-like motif (Y78A and K88A) double mutant plasmid and named Myc-C11orf54-2mu. Then we transfected cells with Flag-HIF1A, Flag-HIF1A plus Myc-C11orf54 and Flag-HIF1A plus Myc-C11orf54-2mu and performed new co-IP assay. Result indicates that the KFERQ-like motif mutation of C11orf54 don't affect the interaction between HIF1A and HSC70 and which may be not the interaction domain.

Fig.5 Immunoprecipitation /Immunoblot analysis of the whole-cell lysates (WCLs) and Anti-HA derived from the 293T cells transfected with indicated plasmids for 48 h. IgG served as a negative control.

6- If C11orf54 has high affinity for HSC70. So, why C11orf54 is not degraded by CMA? Authors should test it.

In the CMA process, HSC70 recognizes the target protein with KFERQ motif to form the HSC70-substrate complex, then the complex interacts with the cytosolic tail of LAMP2A, which drives the translocation the target protein into the lysosome for degradation.

In our study, as is shown in Fig.8C, we paste it below, the immunoprecipitation assay indicates C11orf54 may just interact with HSC70 rather than LAMP2A, which doesn't influence the CMA degradation of C11orf54. Furthermore, we also evaluated the

expression of C11orf54 under 6-aminonicotinamide (6-AN, chaperone-mediated autophagy activator) treatment. We found that CMA activation have no influence on the expression of C11orf54.

Fig.6 CMA activation have no influence on the expression of C11orf54.

A Immunoprecipitation /Immunoblot analysis of the whole-cell lysates (WCLs) and Anti-C11orf54 derived from cells. IgG served as a negative control.

B Western blots of the indicated proteins after 6-AN treatment.

7-Even the use of BafA1 and other lysosomal function inhibitors, authors should knock down Lamp2A to block CMA activity and rescue the HIF1A protein levels (decreased by the knocking down of C11orf54).

As recommended, we silenced LAMP2A by siRNA in C11orf54 knockdown cells. As is shown below, block CMA activity by silencing LAMP2A could rescue the protein of HIF1A in C11orf54 knockdown cells, which could further prove C11orf54 mediated CMA degradation HIF1A.

Fig.7 Western blots of the indicated proteins in control, C11orf54 knockdown and C11orf54, LAMP2A double knockdown cells.

8- Finally, in my opinion, the blocking of HSC70-HIF1A interaction by C11orf54 is not a function of the C11orf54 protein. This is a biological effect and should be described in a different way than a "function".

We agree with you and have corrected the description in the revised manuscript.

Reviewer #2 (Remarks to the Author):

Tan and colleagues show that depletion of C11orf54, a biomarker protein found in renal cancers decreases cell viability and that this effect seems to be further potentiated by cisplatin treatment. The authors show that depletion of C11orf54 leads to loss of RAD51 expression which then leads to a HR defect. Mechanistically they demonstrate that C11orf54 could competitively interact with HSC70 to dissociate HIF1A-HSC70 interaction. When C11orf54 is inhibited this leads to an increased association of HSC70 with HIF1A and as a result promotes CMA mediated HIF1A degradation. Increased HIF1A degradation then leads to reduced RRM2 expression and dNTP imbalance, as RRM2 is a transcription target of HIF1A. The authors attribute the dNTP imbalance as the cause of homologous recombination repair deficiency and genome instability. The findings of this study are novel and will be of interest in the field.

We would like to thank reviewer #2 who approves the novelty and significance of our paper and points out those writing errors. As requested, we have corrected the errors and carefully revised the manuscript.

Major points

1-The mechanism of how C11orf54 regulates RRM2 and autophagy seems interesting but feel that there are several different mechanisms operating in response to loss of C11orf54 in this study. I think that each specific claim (listed in the paper) is experimentally shown. However, connections (causal relationships) among each finding are not well established. Thus, these observations can be interpreted in many different ways, as has been mentioned by the authors. For example how RAD51 expression is regulated by C11orf54 via HIF1A/RRM2 suppression is unclear. Moreover, if HIF1A/RRM2 defects were the causative factors causing loss of viability in C11orf54 depleted cells then overexpressing these should reverse these phenotypes. In addition, the authors do not show that.

1) HIF1A or RRM2 depletion causes loss of viability in these cells and
2) double depletion of C11orf54/HIF1A or C11orf54/RRM2 should be epistatic in terms of the decrease in cell viability if the mechanism is via this axis. These experiments must be carried out if the authors want to reach to the conclusion that the mechanism is via HIF1A/RRM2.

Moreover, like C11orf54 depletion, does HIF1A or RRM2 depletion also cause cisplatin sensitivity or is this a separate mechanism isolated to C11orf54 loss. Authors suggest a model which they think most likely explains the mechanism of how C11orf54 promotes DNA repair, but I think it includes too many hypothesis, which haven't been fully proven.

As suggested, we constructed HIF1A and RRM2 siRNA. According to the CCK8 results, we found that HIF1A and RRM2 depletion both inhibit cell viability.

Furthermore, double depletion of C11orf54/HIF1A and C11orf54/RRM2 also inhibit cell viability, but without synergistic effect. These suggest that C11orf54 is on the same axis as HIF1A/RRM2. According to the CCK8 results, HIF1A and RRM2 depletion also cause cisplatin sensitivity. Furthermore, we found that RRM2 overexpression in C11orf54 knockdown cells could partially rescue proliferation.

Fig.1

A-B. Western blot and CCK8 results of C11orf54 and HIF1A double knockdown.
C-D. Western blot and CCK8 results of C11orf54 and RRM2 double knockdown.
E. CCK8 assay shows the cisplatin sensitivity under each condition.
F. CCK8 results RRM2 overexpression in C11orf54 knockdown cells.

Specific points

2-Figure 2. Authors use two cancer cell lines to investigate the role of C11orf54, one kidney and one liver cancer. To demonstrate that depletion of C11orf54 causes cancer cell inhibition the authors must demonstrate the

effect in a larger panel of cell lines. In particular the authors should use more renal cancer cell lines since C11orf54 has been shown to be a biomarker in this disease setting. What is the effect in normal cells? The authors should investigate if there is a relationship between high C11orf54 expression in renal cancer cell lines vs low expression and cell inhibition in response to C11orf54 depletion. In addition, using TCGA or another publically available dataset show how C11orf54 is a biomarker in cancer? show a figure illustrating expression across tumour types, as supplementary data.

By analyzing the expression profile of C11orf54 from TCGA database, we find that C11orf54 is down-regulated in most cancers especially in kidney carcinoma (KICH, KIRP). As suggested, we silenced C11orf54 in human normal kidney cell line HK-2 and carcinoma cell line Caki-1. Interestingly, we found that C11orf54 depletion in Caki-1 could inhibit the proliferation, while there was no effect on normal HK-2 cells. Besides, C11orf54 knockdown in Caki-1 rather than HK-2, could decrease the expression of HIF1A and RRM2.

Fig.2

A-B. CCK8 and Western blot results of pLKO.1 and shC11orf54 in Caki-1 cells.

C-D. CCK8 and Western blot results of pLKO.1 and shC11orf54 in HK-2 cells.

E. The expression of C11orf54 in several cancer tissue samples based on TCGA data from the GEPIA website

3-The significant decrease in EdU incorporation in the C11orf54 depleted cells, suggests significant disturbance in S-phase. The authors should look at

replication dynamics using fibre analysis in C11orf54 depleted cells vs wild type cells. Its very likely that C11orf54 has a role in replication since the cells are sensitive to replication blocking agents. Is the localization of C11orf54 altered in response to DNA damage – for example is it now localized to the nucleus given the previous reports?

As suggested, we conducted the DNA fiber assay to monitor the dynamics of replication forks. We labeled pLKO.1 and C11orf54 cells with CldU (red) and IdU (green) sequentially, followed by track length analysis. We found that the track length in shC11orf54 group was shorter than that in pLKO.1 group, which meant loss of C11orf54 inhibited the progress of replication forks.

Furthermore, we detected C11orf54 location after DNA damage treatment by nuclear/cytosolic-fractionation assay and immunostaining experiment. We found that C11orf54 was still located in cytoplasm after Cisplatin treatment, rather than nucleus. These data indicate that DNA damage can't alter the subcellular location of C11orf54.

Fig.3

*A-B. Representative images of DNA fiber and statistical result of track length in pLKO.1 and shC11orf54 cells. Scale bar = 5 µm, ***p < 0.001.*

C. Nuclear/cytosol fractionation assay shows subcellular localization of C11orf54 upon CT and Cisplatin treatment. GAPDH was used as a cytoplasm (Cyto) marker and H3 as a marker of the nucleus (Nuc).

D. Representative immunostaining images confirm the subcellular localization of C11orf54 upon CT and Cisplatin treatment. Scale bar = 10 µm

4-Figure 3. The authors state that the DNA damage signalling is ATM

dependent, have they tested ATR as well since the signalling was only partially restored in response to ATM inhibitor?

We appreciate reviewer #2 for raising this important question. In the DNA damage pathway, ATM and ATR are the most upstream DDR kinases, which can activate a second wave of phosphorylation of CHK1 and CHK2. Although ATM and ATR have distinct specificities, they may cross talk with each other (PMID: 17124492). On the one hand, ATM and ATR may directly phosphorylate each other. On the other hand, ATM and ATR may provide functional redundancy during the DDR. Even in the absence of ATM, slow resection of DSBs still activates ATR (PMID: 19444312). Similarly, when ATR signaling is compromised, ATM is activated by collapsed replication forks (PMID: 19049966).

In our study, we found that loss C11orf54 inhibited the phosphorylation of ATR. We speculate that knockdown of C11orf54 may result in the crosstalk between ATM and ATR. Loss C11orf54 inhibits ATR pathway and activates ATM pathway.

Fig.4 Western blot results of p-ATM and p-ATR in pLKO.1 and shC11orf54 cells.

5-Figure 4. The authors must look at RAD51 foci and chromatin association of RAD51. Does loss of C11orf54 depletion effect RAD51 at the protein or mRNA level?

As is shown in Fig4a, loss C11orf54 inhibited the RAD51 expression at the protein. In addition, we conducted the qPCR experiment and found C11orf54 knockdown didn't affect the mRNA level of RAD51.

As suggested, we preformed immunostaining experiment to observe the RAD51 foci. The results showed that loss of C11orf54 significantly inhibited RAD51 foci in the nucleus.

Fig.5

A-B Western blots (A) and qPCR results (B) of the Rad51 in pLKO.1 and C11orf54 knockdown cells

*C-D Representative images of Rad51 foci (D) and quantitative results (C) in control and C11orf54 knockdown cells at the indicated time point after the cisplatin treatment (n = 10 independent cells; data are presented as mean values ± SD; **p < 0.01). Scale bar = 10 μm.*

6-If dNTP depletion via a defect in HIF1A/RRM2 is the cause of the reduction in RAD51 expression/HR frequency, does dNTP supplementation rescue RAD51 levels back to wild type in C11orf54 depleted cells?

According to the western blot results, dNTP supplementation can't rescue the expression of RAD51.

Fig.6 Western blots of the Rad51 after 100μM and 200μM dNTP treatment for 24 hrs.

7-Can you rescue cisplatin sensitivity by supplementing C11orf54 depleted cells with dNTP?

According to the CCK8 results, supplementing with dNTP could partially rescue cisplatin sensitivity in medium cisplatin concentration.

Fig.7. CCK8 assay shows the cisplatin sensitivity by supplementing with dNTP.

8-In addition, have the authors tested PARPi in their cells since PARPi induce lesions directly engage HR for repair? This should be tested. Its possible that HR frequency is effected due to a significant disturbance in S-phase in C11orf54 depleted cells.

As suggested, we preformed flow cytometry to test the HR repair activity. As is shown below, knockdown of C11orf54 suppressed HR repair activity under normal circumstance. When treating with PARP inhibitor Olaparib, the HR repair efficiency was inhibited significantly.

Fig.8. Flow cytometric analysis (A) and quantitative results (B) of the GFP-positive cells in pLKO.1 and C11orf54 knockdown cells upon CT and Olaparib treatment.

9-Figure 5 Show a table of differentially expressed genes in the cell lines shown in this figure. Did RAD51 score in the RNAseq data? What about RRM2?

EnsemblGene	GeneSymbol	plko-1	plko-2	shC11orf54-1	shC11orf54-2
ENSG00000051180	RAD51		510	413	603
	RRM2				715

EnsemblGene	GeneSymbol	plko-1	plko-2	shC11orf54-1	shC11orf54-2
ENSG00000171848	RRM2		2605	2510	1183
					1258

By analyzing the RNAseq data, we found that there was no differential expression about Rad51 between plko and shC11orf54, which matched with our mRNA and protein results. RRM2 was one of the most differential expression gene and was down expression in shC11orf54 group.

10-Figure 6. The authors need to clearly justify how they selected to look at the DNA repair factors in Figure 6A. There are many more DNA repair genes why were these particular ones selected? Did they come up in the RNAseq data? Why did you not also assess RAD51 here?

In Fig.4 we have demonstrated that loss of C11orf54 inhibited HR repair, so to further investigate how C11orf54 regulates DNA damage repair, we measured the mRNA levels of genes involved in DNA repair as previous study (PMID: 18842680)

Figure quote from PMID: 18842680)

Thus, we select these genes to test. As our data showed that most DNA repair-related genes are not changed in C11orf54 knockdown cells. Only RRM2 was down-regulated both in mRNA and protein levels, which was in accord with our RNA-seq data. We tested the expression of Rad51 and found that only the protein but not the mRNA level down-regulated. We have added the qPCR data of Rad51 in Fig6A.

11-Does RRM2 depletion cause the same level of cell inhibition/DNA damage

as depleting C11orf54, is co-depletion of RRM2/C11orf54 epistatic in terms of cell inhibition? What about HIF1A?

As suggested, we constructed HIF1A and RRM2 shRNA. According to the CCK8 results, we found that HIF1A and RRM2 depletion both inhibit cell viability.

Furthermore, double depletion of C11orf54/HIF1A and C11orf54/RRM2 also inhibit cell viability, but without synergistic effect

Fig.9 Western blot and CCK8 results of C11orf54 and HIF1A/RRM2 double knockdown.

12-Figure 7. Does BafA1 treatment in C11orf54 depleted cells rescue the cell viability observed in response to cisplatin treatment?

As is shown below, pretreatment with BafA1 could partially restore the cell viability after C11orf54 knockdown in response to cisplatin treatment.

Fig.10. CCK8 assay shows the cell viability by pretreatment with BafA1.

There are many textual errors found throughout the paper including in the schematic figure of the mechanism (Figure 9) – should say “Absence of C11orf54” not “Absentent of C11orf54” and “HIF1A degradation” not “HIF1A

degradation". The text throughout the paper must be thoroughly checked by the authors.

As required, we have corrected the textual mistakes in the revised figure.

Minor point

The absence of something cannot promote – rephrase so that it says causes rather than C11orf54 depletion promotes throughout the text

Rephrase line 179-180 about ATM and ATR

Please rephrase line 233-236

Figure 7 instead of using promotes please use causes

As required, we have rephrased the tense in the revised manuscript.

Reviewers' comments:

Reviewer #1 (Remarks to the Author):

Authors have addressed the suggested comments.

Reviewer #2 (Remarks to the Author):

The authors addressed my main concerns with the paper. However, one comment that remains is that the authors still state that RRM2 and dNTP imbalance as a result of C11orf54 loss leads to a HR defect "eventually leading to homologous recombination repair deficiency". They cannot make this claim without proving that dNTP supplementation can actually rescue this phenotype. In fact dNTP supplementation does not rescue RAD51 expression in C11orf54 knockout cells as per their rebuttal. This claim needs to be removed from the manuscript. Stating that there must be an alternative mechanism effecting HR.

There are still many textual errors throughout the paper. I have listed a few below but this does not include all errors. The authors need to correct these before publication.

Some are listed below;

30 promoted HSC70 binds HIF1A should be "HSC70 binding to HIF1A"

31 C11orf54 knockdown-mediated HIF1A degradation reduced the
32 transcription of ribonucleotide reductase regulatory subunit M2 (RRM2), resulting in
33 dNTP pool imbalances, eventually leading to homologous recombination repair
34 deficiency and genome instability.

Conclusion is still incorrect that dNTP imbalances cause HR repair deficiency and supplementation with dNTP should rescue – the authors need to include this in their model and conclusion that it could be via another mechanism not involving dNTP.

138 C11orf54 deficiency suppressed cell proliferation and promoted
139 apoptosis. No full stop

157 V(FITC)/Propidium iodide (PI) double staining showed that Cisplatin-induced early
158 and late phase apoptosis was reduced in the C11orf54 knockdown cells (Fig 2K-L). It should say
increased not reduced

Knockdown of C11orf54 promotes DNA damage via suppression of
169 homologous recombination. No full stop

indicating that C11orf54 may
173 involve in genome stability or DNA damage. Should say may be involved in genome stability

202 in nucleus (Fig 4C-D), which indicating that C11orf54 may be involved in homologous
203 recombination. Should say indicated

We

220 identified 708 significant differential expression genes (303 up-regulated genes and 405
221 down-regulated genes) in C11orf54 knockdown cells, Should say differentially expressed genes

239 Next, we explored which downstream target of HIF1A was involved in dysregulated
240 DNA repair in C11orf54 knockdown cells. Rephrase this sentence

274 suggesting that knockdown of C11orf54 repressed HIF1A may not via the regulation of
275 HIF1A stabilization. Should say may not be

Wortmannin and 3-MA treatment, which repressed 288 the early phase of autophagy, lose
289 the ability to restore the level of HIF1A in C11orf54 knockdown cells (Fig S5B-C). Rephrase

291 These results suggest that knockdown of C11orf54 reduces HIF1A
292 may through the lysosome participated in late phase of autophagy. Rephrase

Since HIF1A contained a KFERQ-like motif, which could be recognized and binding
318 by HSC70(48), we suppose that C11orf54 competitive interact with HSC70, which
319 blocks the interaction between HIF1A and HSC70. Rephrase

358 in KICH, KIRP and SARC (Fig S7). In our study, C11orf54 knockdown could cause
359 DNA damage and inhibit proliferation (Fig 2-3). These suggest that the c11orf54
360 expression is decreased in cancer tissue through an unknown mechanism, which may
361 be a feedback loop to block the tumor cell survival. Describe KICH, KIRP and SARC

Pages 354-361 different font used in main text

HSC70 and LAMP2A(20). Early studies believed macroautophagy and chaperone385
mediated autophagy are complementary in proteins for eliminating(53). Rephrase

Point-by-point response

We appreciate the reviewers for considering the strengths of our work and for their valuable advice and suggestions for improving this manuscript. We have tried our best to address these points by conducting new experiments and revising the manuscript. Below are our point-by-point responses (*blue italic type*) to the reviewers' comments.

=====

Reviewer #1 (Remarks to the Author):

Authors have addressed the suggested comments.

We would like to thank reviewer #1 for the comments that helped us improve our manuscript.

Reviewer #2 (Remarks to the Author):

The authors addressed my main concerns with the paper. However, one comment that remains is that the authors still state that RRM2 and dNTP imbalance as a result of C11orf54 loss leads to a HR defect “eventually leading to homologous recombination repair deficiency”. They cannot make this claim without proving that dNTP supplementation can actually rescue this phenotype. In fact dNTP supplementation does not rescue RAD51 expression in C11orf54 knockout cells as per their rebuttal. This claim needs to be removed from the manuscript. Stating that there must be an alternative mechanism effecting HR.

We would like to thank reviewer #2 for this important suggestion. We have removed the claim that dNTP imbalance leads to HR repair deficiency and stated that there must be an alternative mechanism effecting HR in the discussion part. We also revised the schematic model to show that C11orf54 regulates DNA repair through HIF1A/RRM2 and Rad51.

There are still many textual errors throughout the paper. I have listed a few below but this does not include all errors. The authors need to correct these before publication.

Some are listed below;

Thanks for pointing out these textual errors. As suggested, we have tried our best to correct the errors below and some other textual and grammatical mistakes in the revised manuscript.

30 promoted HSC70 binds HIF1A should be “HSC70 binding to HIF1A”

31 C11orf54 knockdown-mediated HIF1A degradation reduced the
32 transcription of ribonucleotide reductase regulatory subunit M2 (RRM2),
resulting in
33 dNTP pool imbalances, eventually leading to homologous recombination
repair
34 deficiency and genome instability.

Conclusion is still incorrect that dNTP imbalances cause HR repair deficiency
and supplementation with dNTP should rescue – the authors need to include
this in their model and conclusion that it could be via another mechanism not
involving dNTP.

138 C11orf54 deficiency suppressed cell proliferation and promoted
139 apoptosis. No full stop

157 V(FITC)/Propidium iodide (PI) double staining showed that Cisplatin-
induced early
158 and late phase apoptosis was reduced in the C11orf54 knockdown cells
(Fig 2K-L). It should say increased not reduced

Knockdown of C11orf54 promotes DNA damage via suppression of
169 homologous recombination. No full stop

indicating that C11orf54 may
173 involve in genome stability or DNA damage. Should say may be involved
in genome stability

202 in nucleus (Fig 4C-D), which indicating that C11orf54 may be involved in
homologous
203 recombination. Should say indicated

We
220 identified 708 significant differential expression genes (303 up-regulated
genes and 405
221 down-regulated genes) in C11orf54 knockdown cells, Should say
differentially expressed genes

239 Next, we explored which downstream target of HIF1A was involved in
dysregulated
240 DNA repair in C11orf54 knockdown cells. Rephrase this sentence

274 suggesting that knockdown of C11orf54 repressed HIF1A may not via the regulation of
275 HIF1A stabilization. Should say may not be

Wortmannin and 3-MA treatment, which repressed 288 the early phase of autophagy, lose
289 the ability to restore the level of HIF1A in C11orf54 knockdown cells (Fig S5B-C). Rephrase

291 These results suggest that knockdown of C11orf54 reduces HIF1A
292 may through the lysosome participated in late phase of autophagy. Rephrase

Since HIF1A contained a KFERQ-like motif, which could be recognized and binding
318 by HSC70(48), we suppose that C11orf54 competitive interact with HSC70, which
319 blocks the interaction between HIF1A and HSC70. Rephrase

358 in KICH, KIRP and SARC (Fig S7). In our study, C11orf54 knockdown could cause
359 DNA damage and inhibit proliferation (Fig 2-3). These suggest that the c11orf54
360 expression is decreased in cancer tissue through an unknown mechanism, which may
361 be a feedback loop to block the tumor cell survival. Describe KICH, KIRP and SARC

Pages 354-361 different font used in main text

HSC70 and LAMP2A(20). Early studies believed macroautophagy and chaperone385
mediated autophagy are complementary in proteins for eliminating (53). Rephrase